# The structural basis of function and regulation of neuronal cotransporters NKCC1 and KCC2

Sensen Zhang [1,7], Jun Zhou [1,7], Yuebin Zhang [2,7], Tianya Liu [1,7], Perrine Friedel [3,7], Wei Zhuo [1], Suma Somasekharan [3], Kasturi Roy [3], Laixing Zhang [1], Yang Liu[1], Xianbin Meng[4], Haiteng Deng[4], Wenwen Zeng [5], Guohui Li [2✉], Biff Forbush [3✉] & Maojun Yang [1,6✉]

NKCC and KCC transporters mediate coupled transport of $Na^+ + K^+ + Cl^-$ and $K^+ + Cl^-$ across the plasma membrane, thus regulating cell $Cl^-$ concentration and cell volume and playing critical roles in transepithelial salt and water transport and in neuronal excitability. The function of these transporters has been intensively studied, but a mechanistic understanding has awaited structural studies of the transporters. Here, we present the cryo-electron microscopy (cryo-EM) structures of the two neuronal cation-chloride cotransporters human NKCC1 (SLC12A2) and mouse KCC2 (SLC12A5), along with computational analysis and functional characterization. These structures highlight essential residues in ion transport and allow us to propose mechanisms by which phosphorylation regulates transport activity.

[1] Ministry of Education Key Laboratory of Protein Science, Tsinghua-Peking Center for Life Sciences, Beijing Advanced Innovation Center for Structural Biology, School of Life Sciences, Tsinghua University, Beijing, China. [2] State Key Laboratory of Molecular Reaction Dynamics, Dalian Institute of Chemical Physics, Chinese Academy of Sciences, Dalian, China. [3] Department of Cellular and Molecular Physiology, Yale University School of Medicine, New Haven, CT, USA. [4] MOE Key Laboratory of Bioinformatics, School of Life Sciences, Tsinghua University, Beijing, China. [5] Center for Life Sciences, Institute for Immunology and School of Medicine, Tsinghua University, Beijing, China. [6] School of Pharmacy, Tongji Medical College, Huazhong University of Science and Technology, Wuhan, China. [7] These authors contributed equally: Sensen Zhang, Jun Zhou, Yuebin Zhang, Tianya Liu, Perrine Friedel. ✉email: ghli@dicp.ac.cn; biff.forbush@yale.edu; maojunyang@tsinghua.edu.cn

Cation-chloride cotransporters (CCC) are central to the regulation of intracellular Cl⁻ concentration and have key physiological roles in neuronal excitability, trans-epithelial salt and water movement, and regulation of cell volume[1–8]. In humans, there are two broadly distributed subfamilies: the Na⁺(K⁺)-coupled transporters NKCC1, 2 and NCC (*SLC12A1-3*) and the K⁺-coupled KCC1-4 (*SLC12A4-7*). The NKCCs and NCC utilize the energy of the Na⁺ gradient generated by Na⁺/K⁺ ATPase[5] to accumulate cellular Cl⁻ above its electrochemical equilibrium while KCCs utilize the outwardly directed K⁺ gradient to extrude Cl⁻. These transporters generate electrochemical Cl⁻ gradients across the neuronal plasma membrane while channels dissipate them, and these interactions set the polarity and driving force of GABA$_A$ receptor-mediated chloride currents[4,9]. The CCCs are strictly electroneutral transporters, NKCCs transporting two Cl⁻ ions coupled to the movement of one Na⁺ and one K⁺, whereas NCC and KCC couple the movement of one Cl⁻ ion and either Na⁺ or K⁺ respectively. From the kinetic analysis, there is strong evidence that in NKCC one Na⁺ and one Cl⁻ ion bind before K⁺ binds, and it also appears that Na⁺ is released first on the inside[10,11]; similarly, in KCC the kinetic evidence points to the ordered binding of first Cl⁻ and then K⁺ from the outside[12]. NKCC is inhibited by the loop diuretics bumetanide and furosemide and KCC is inhibited by the same compounds but at higher concentrations[13–16]—NCC is evolutionarily close to NKCC but is inhibited by thiazide diuretics and not by bumetanide[5,17].

NKCC1 and KCC2 are expressed in mammalian central neurons, in which a developmental hyperpolarizing shift in the action of GABA takes place during neuronal maturation because of an increase in the functional expression of KCC2[9]. Thus, by raising [Cl⁻]$_i$, NKCC1 is responsible for the depolarizing and sometimes excitatory GABA actions in immature central neurons, and also for the GABA-mediated (but functionally inhibitory) depolarization in adult sensory neurons, while the CNS-specific KCC2 determines the inhibitory GABA response in mature neurons by lowering [Cl⁻]$_i$[18]. All CCCs are feedback-regulated by the concentration of intracellular Cl⁻—this is mediated by the WNK (kinase with no lysine)-SPAK/OSR1 (SPS1-related proline/alanine-rich kinase/oxidative stress-responsive kinase 1) signaling pathway[19]. Any change in [Cl⁻]$_i$ is directly sensed by WNK kinases which phosphorylate SPAK/OSR1 kinases; in turn, SPAK/OSR1 directly phosphorylates threonine residues in the N terminus of NKCC to activate the transporter and promote Cl⁻ entry and in the C-terminal domain of KCCs to inactivate that transporter. In the case of KCC2, membrane trafficking is also tied to phosphorylation of the protein and is thus an important component of KCC2 regulation, as is PKC-mediated phosphorylation of a serine in the C terminus[20]; similarly, regulation of NKCC2 by phosphorylation has been shown to involve both changes in intrinsic transport activity and trafficking[21]. A structural explanation for the coupling of phosphorylation to CCC transport activity is still not available, but in NKCC the mechanism appears to involve the movement of the C terminus[22] and a change in the architecture of TMs 10 and 12[23].

The critical role of NKCC1 and KCC2 in setting neuronal [Cl⁻]$_i$ and cell volume leads to a wide range of neuropathology when the CCCs or the WNK-SPAK/OSR1 pathway are perturbed. Genetic defects resulting in altered expression or function of NKCC1, KCC2, WNK, or SPAK/OSR1 have been tied to the pathogenesis of seizures, neuropathic pain, cognitive impairment, cerebral edema, and neurodegeneration[8,24,25]. Changes in Cl⁻ regulation are likely to have an important role in the molecular etiology of a number of psychiatric and neurological diseases— since some of the deleterious effects on neuronal development and function have been connected with elevated [Cl⁻]$_i$ and/or cell volume and increased strength of GABA signaling, a number of recent efforts have focused on inhibition of NKCC1, inhibition of WNK-SPAK/OSR1, or on finding agonists to increase KCC2 transport activity and membrane trafficking[19,26]. The promise of curing or treating these life-changing and life-threatening neuropathies drives the current quest for a full understanding of the CCC regulatory mechanisms.

Although the properties of ion translocation, regulation, and inhibitor interactions of Na⁺ and K⁺ coupled CCCs have been extensively investigated[3–5,7,17,27–33] until recently the molecular mechanisms have remained elusive due to the lack of structural information. Starting a decade ago, crystallographic structures of prokaryotic amino acid transporters AdiC, ApCT, and GkApcT in the same APC (amino acid-polyamine-cotransporter) superfamily enabled informative homology modeling efforts[29,34–36] with moderate expected fidelity but little information as to ion-binding sites or regulation. With the rise of cryo-electron microscopy, structures of other APC family members have been reported[37–40], including one Na⁺-coupled amino acid transporter AgcS[37]. Recently, four CCC structures, including the DrNKCC1[40], human NKCC1(K289N_G351R)[41], human KCC1[39], and mouse KCC4[42] have been reported at 2.9–3.7 Å resolution, and these have revealed putative ion-binding sites and key residues likely involved in transport conformational change.

Here, using single-particle cryo-electron microscopy (cryo-EM), we report the structures of human NKCC1 (hNKCC1a) and mouse KCC2 (mKCC2a) at 3.5 and 3.8 Å resolution, respectively. Our examination of human NKCC1 in an inward-open inactive state reveals conservation of key features in common with the reported NKCC structures[40,41] and we combine functional studies with molecular dynamics (MD) simulations to reveal aspects of the structure such as ion binding and translocation, the identity of gating residues, and dimerization. The mKCC2 structure is in a similar inward-open inactive state but with an inhibitory N-terminus peptide in place to block the translocation pore, and with a very different dimer architecture compared to KCC1— together these exciting structures allow us to propose models for the regulatory mechanisms of CCCs and to gain insight into the defects in known human mutations.

## Results

**Structure determination of hNKCC1 and mKCC2.** To gain insight into the architecture of mammalian cation-chloride cotransporters (NKCC, KCC, and NCC), we screened candidate CCC transporters from different species, including *Homo sapiens* and *Mus musculus*. Based on biochemical stability and structural homogeneity, we identified hNKCC1a and mKCC2a as the top two candidates suited for high-resolution studies (Supplementary Fig. 1)—these candidate proteins were purified from mammalian cells through Strep-tag II affinity chromatography and reconstituted in digitonin buffer for cryo-EM study. We were able to determine the structures of hNKCC1 and mKCC2 at 3.5 and 3.8 Å resolution, respectively, and these were of sufficient quality to build models for major parts of these proteins (Fig. 1, Supplementary Figs. 1–3 and Table 1). Similar to DrNKCC1[40], hNKCC1(K289N_G351R)[41], and hKCC1[39], hNKCC1 and mKCC2 structures both exhibit dimeric architecture in a domain-swap configuration with a dimer volume of 140 Å × 110 Å × 65 Å (Fig. 1 and Supplementary Fig. 3a). This dimeric arrangement is consistent with extensive biochemical evidence, including cross-linking and FRET studies[43,44]. The final model of hNKCC1 was primarily focused on the transmembrane and extracellular regions from the high-resolution map (Fig. 1a, b), although we were also able to determine the overall structure of hNKCC1 including the dimeric C termini from a medium resolution map

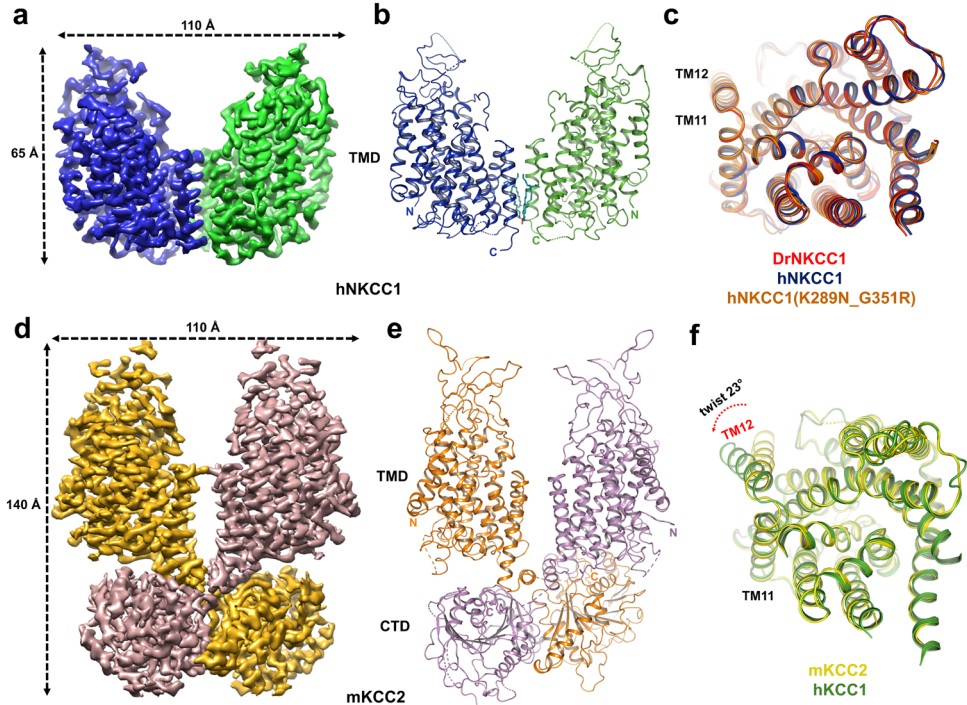

**Fig. 1 Structures of hNKCC1 and mKCC2. a** Cryo-EM map of the high-resolution hNKCC1 from a side view with each subunit color-coded (blue and green). **b** Ribbon representation of the high-resolution hNKCC1 from a side view with each subunit color-coded (blue and green). **c** The TM-domain of the DrNKCC1 (red), previous reported hNKCC1(K289N_G351R) (brown), and the current hNKCC1 (blue) NKCC1 single monomers are superimposed. Top view (from the extracellular side). **d** Cryo-EM map of the dimeric mKCC2 from a side view with each subunit color-coded (yellow and brown). **e** Ribbon representation of the dimeric mKCC2 from a side view with each subunit color-coded (yellow and brown). **f** Superimposition of single monomers of hKCC1 (green) and mKCC2 (yellow) in the TMD demonstrates good alignment (RMSD of 0.69 Å over 341 Cα atoms) except for the TM12 helix which is tilted 23° in hKCC1 compared to mKCC2.

(Supplementary Fig. 3a, b). We successfully built most of the mKCC2 structure except for relatively disordered regions in the N terminus and extracellular and intracellular loops (Fig. 1d, e and Supplementary Fig. 3c).

**Overall structure of hNKCC1 and mKCC2.** Like other members of the APC superfamily[34–40,45,46], hNKCC1 and mKCC2 exhibit internal symmetry (Supplementary Figs. 3d, e, and 4), with TM1-TM5 related to TM6-TM10 by a pseudo-twofold symmetrical axis perpendicular to the lipid membrane in a "LeuT fold"[47]. TM1 and TM6, oriented antiparallel to each other, are discontinuous helices, which are broken in the center of the transporter. Intracellular loop 1 (ICL1) intervening TM2 and TM3, is a highly conserved region among all CCC family members and likely has an important role in ion transport and regulation[5,29,48,49] (Supplementary Fig. 5). Our structures of hNKCC1 and mKCC2 exhibit these structural features in an inward-open configuration, like the reported structures of DrNKCC1[40], hNKCC1 (K289N_G351R)[41], mKCC4[42], and hKCC1[39] (Fig. 1 and Supplementary Fig. 3), with a pore in the middle of the TM domain that is open only to the intracellular medium.

The transport activity of both NKCCs and KCCs is controlled by the phosphorylation state of the transport proteins—in the case of NKCCs (and NCC) transport activity occurs only when the N terminus is phosphorylated[32], whereas in KCCs the dephosphorylated protein is the active state[50,51]. We have strong evidence that both of our structures represent inactive states. In the case of hNKCC1, we analyzed phosphorylation of the N terminus using a phospho-specific antibody—as shown in Supplementary Fig. 6a, there is no detectable phosphorylation of the protein as prepared for cryo-EM, and from extensive previous reports[32,52,53], we, therefore, know that this protein

must be inactive. In the case of mKCC2, using mass spectrometry we confirmed phosphorylation at the two sites in the C terminus (T929 and T1029) (Supplementary Fig. 6b, c) whose phosphorylation appears necessary and possibly sufficient to inactivate KCC transport function; additional evidence will be seen below from the blocked pore in the mKCC2 structure.

The activating phosphorylation sites of NKCC1 are in the N-terminal domain[32]—this region is predicted to be largely disordered and indeed the N terminus is absent from both NKCC1 structures[40,41]. The phosphorylation sites in mKCC2 that deactivate the transport protein (T929 and T1029)[50] are in a large loop of the C terminus intervening β9 and α7—the ends of this loop are highlighted in Supplementary Fig. 7, but the majority of the loop is unresolved in our mKCC2 structure. This region also includes the mKCC2-specific "isotonic domain" (ISO domain, 1043–1057), which has been shown to confer constitutive isotonic transport[54].

Our structure of hNKCC1 is extremely similar to the structure of DrNKCC1 and hNKCC1(K289N_G351R) that has been recently reported[40,41], as illustrated in Fig. 1c and Supplementary Fig. 3a, b, d. This demonstrates a high level of evolutionary conservation and assures that both structures are in the same inward-open inactive conformation. Like in DrNKCC1[40] and hKCC1[39], we note densities within the cleft at the dimer interface and we attribute these to four lipid molecules (Supplementary Fig. 8a) with lipid tails in the hydrophobic cavity contributed by TM residues (Supplementary Fig. 8b), and hydrophilic heads facing intracellularly (Supplementary Fig. 8c). The carboxyl head groups of two of these lipids interact with H695 from TM10 and Y751 from TM12, thereby forming a triangle interaction network that presumably stabilizes TM11-12 and the dimeric architecture (Supplementary Fig. 8d).

**Table 1 Cryo-EM data collection, refinement, and validation statistics.**

|  | hNKCC1 (EMD-30542) (PDB-7D10) | mKCC2 (EMD-30543) (PDB-7D14) |
|---|---|---|
| Data collection and processing |  |  |
| Magnification | 105,000 | 105,000 |
| Voltage (kV) | 300 | 300 |
| Electron exposure ($e^-$/Å$^2$) | 50 | 50 |
| Defocus range (μm) | −1.5 ~ −2.5 | −1.5 ~ −2.5 |
| Pixel size (Å) | 1.091 | 1.091 |
| Software | RELION-3.0 | RELION-3.0 |
| Symmetry imposed | C2 | C2 |
| Initial particle images (no.) | 622,382 | 597,572 |
| Final particles images (no.) | 80,568 | 66,684 |
| Map resolution (Å) | 3.52 | 3.8 |
|   FSC threshold | 0.143 | 0.143 |
| Local map resolution range (Å) | 4.5–2.5 | 4.6–2.7 |
| Refinement |  |  |
| Software | PHENIX 1.14 | PHENIX 1.14 |
| Model resolution (Å) | 3.6 | 3.8 |
|   FSC threshold | 0.5 | 0.5 |
| Map sharpening $B$ factor | −180 | −142 |
| Model composition |  |  |
|   Non-hydrogen atoms | 6440 | 11,664 |
|   Protein residues | 894 | 1700 |
|   Ligand | 4 | 0 |
| $B$ factors (Å$^2$) |  |  |
|   Protein | 48.19 | 62.81 |
|   Ligand | 26.71 | – |
| R.m.s deviations |  |  |
|   Bond length (Å) | 0.007 | 0.007 |
|   Bond angles (°) | 1.375 | 1.286 |
| Validation |  |  |
|   MolProbity score | 1.79 | 1.90 |
|   Clashscore | 4.83 | 5.97 |
|   Poor rotamers (%) | 0.00 | 0.4 |
| Ramachandran plot |  |  |
|   Favored (%) | 90.43 | 89.23 |
|   Allowed (%) | 9.57 | 10.77 |
|   Disallowed (%) | 0.00 | 0.00 |

The structures of mKCC2 and hKCC1[39] are very similar in the transmembrane domain, with an RMSD of 0.69 Å over 341 Cα atoms in an alignment of TM1 to TM11 (Fig. 1f)—this is consistent with the high degree of conservation of the transport motif (Supplementary Fig. 5) and an inward-open transport configuration in both KCCs. There are however striking differences between the two KCC structures in (a) the presence of an inhibitory N-terminal peptide in mKCC2 (Fig. 2a), (b) a large displacement of the KCC1 dimer interface compared to that seen in the KCC2 structure (Supplementary Fig. 3e), (c) an angular displacement of TM12 in the KCC1 vs KCC2 structures (Fig. 1f), and (d) the presence of a well-ordered C-terminus dimer in mKCC2 but not in hKCC1 (Fig. 1e). We propose that these differences are all related and most likely represent the difference between the inactive and active status of the KCC transporter.

**N-terminus inhibitory peptide in mKCC2 structure.** Within the TM region of mKCC2, we noted a helix-loop density that we were able to assign to residues 85–108 in the N terminus, lying in a cleft between TMs 1, 5, 6, and 8 (Fig. 2a, b). Residues T92 and

Q96 from the N terminus interact with R443 and S444 from TM6b, while the additional N-terminus residues N93 and E102 interact with Q524 and R538 from TM8 (Fig. 2c). The presence of this bound N-terminus peptide is a marked finding since at this position it is expected to completely inhibit function: in addition to the inhibitory interactions with critical TMs 6 and 8, the peptide completely blocks access to the otherwise-inward-open pore. We propose therefore that binding of this inhibitory peptide is the final step in the deactivation mechanism triggered by KCC phosphorylation. Sequence alignment of the four KCC family members reveals strong conservation of the interacting residues among the KCCs both for the N-terminus residues and those in TMs 6 and 8 (Fig. 2d), suggesting a role for this inhibitory interaction throughout the KCC family.

In a previous report, Casula et al.[55] found that deletion of most of the N terminus of KCC1 resulted in an inactive protein, demonstrating an essential role for at least part of the N terminus. To characterize the proposed inhibitory function of the identified peptide in KCC2, we performed a fluorescence-based influx assay with Tl$^+$ as a congener of K$^{+}$[56,57]. We found Tl$^+$ influx several-fold higher in mKCC2-transfected than in mock-transfected cells, and transporter activity was appropriately inhibited by the KCC inhibitor dihydroindenyloxy alkanoic acid (DIOA) (Supplementary Fig. 9 and Supplementary Data 1). Compared with wild-type mKCC2, a mutant with deletion of residues 85–120 in the N terminus was found to have increased transporter activity (Fig. 2e, f and Supplementary Data 1), strongly supporting an inhibitory role of this N-terminal region in KCC transporters.

**Potential gates in hNKCC1 and mKCC2.** In our inward-open structure of hNKCC1, extracellular access to the pore is blocked by the side chain carbonyl of E389 from TM3 interacting with the side chain amino group of R307 of TM1b and by L671 from TM10 (Fig. 3a)—the narrowest open radius of this potential "gate" is ~1 Å. To test the importance of these residues we examined the transport activity of hNKCC1 mutants. As illustrated in Fig. 3b, mutation of any of these residues resulted in Cl$^-$ transport activity that is <20% of wild-type activity, with changes in R307 and E389 being particularly impactful (Fig. 3b and Supplementary Data 1); on the other hand, mutagenesis of neighboring E670 has little effect on function (Fig. 3b).

At the intracellular end of the inward-open pore, it is noteworthy that R358 from ICL1 interacts with D632 from adjacent TM8 (Fig. 3c). We found that mutagenesis of these two residues did indeed have a dramatic effect on transport activity resulting in 10% or less activity in R358K, D632A, D632E, and R358K/D632E compared to wild-type hNKCC1 (Fig. 3d and Supplementary Data 1). These results confirm an essential role for R358 and D632—interestingly though, the charge at D632 is not absolutely required, as the otherwise conservative D632N mutant retains 25% of the wild-type activity.

Of additional importance at the intracellular end of the pore, the main-chain carbonyls of G349, G350, G509, and side chain of D510 appear to have an important role in modulating cation permeation, and D510 tends to form a hydrogen bond with the neighboring K624 during our MD simulations (Supplementary Fig. 10a, b). We tested the importance of these residues by measurement of transport function—indeed mutagenesis of D510 and K624 reduced the bumetanide-sensitive transport of chloride to less than 25% of the wild-type transport (Fig. 3e and Supplementary Data 1). As for D510, the size of the D510 appears to be important but not the fixed charge—we found that the D510N mutant retained full transport activity. Interestingly, mutants K624C and D510Q exhibit dramatically reduced affinity for bumetanide (Fig. 3e and Supplementary Fig. 10c), possibly

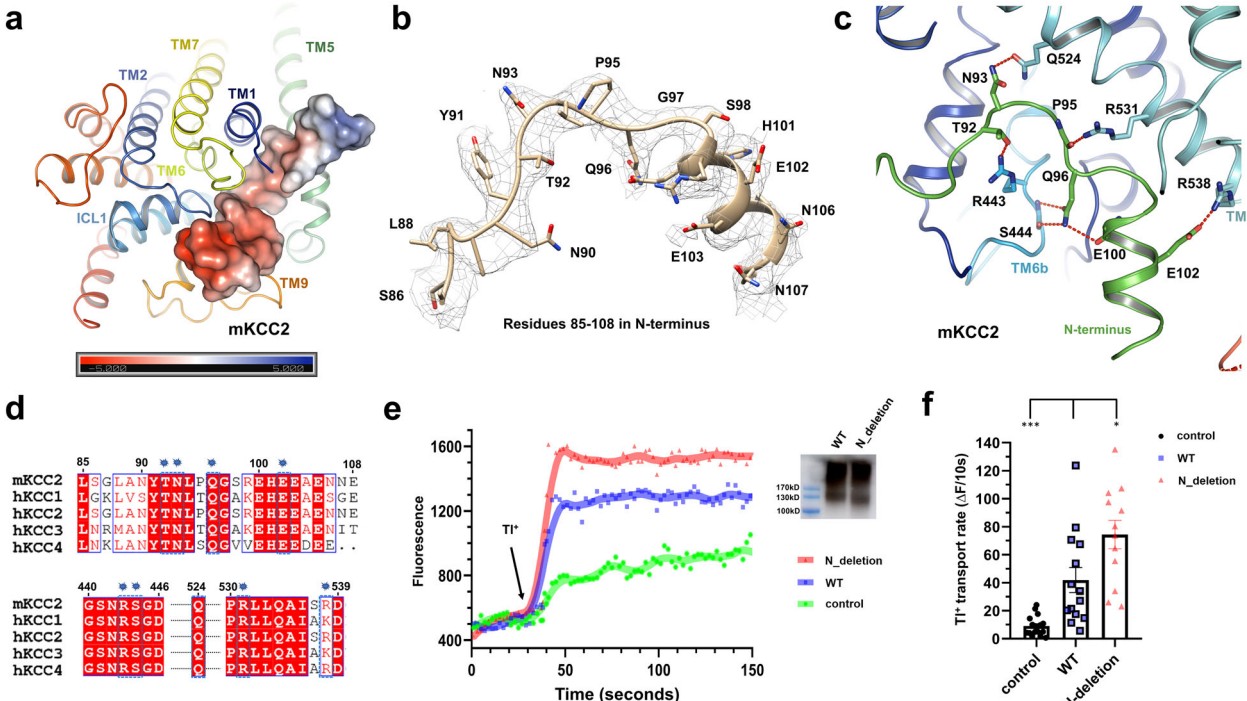

**Fig. 2 Inhibitory N-terminus peptide in mKCC2. a** An N-terminus peptide fragment blocks the entrance to the translocation pore of mKCC2. The peptide is shown as the electrostatic surface potential from -5 to +5 kT/e. **b** Density map of the representative residues 85–108 in N terminus of mKCC2 showing the 10.0 σ contour. Representative atomic model (gold) with side chains was fitted into the cryo-EM density map (gray mesh). **c** Interactions between the N-terminus peptide and TM domains of mKCC2. T92, Q96, N93, P95, and E102 from the N-terminus peptide interact with R443 and S444 from TM6b, and interact with Q524, R531, and R538 from TM8 in mKCC2. **d** Sequence alignment KCCs in the regions of interaction between the N terminus, and of TMs 6 and 8. Numbering is that of mKCC2b. Interacting residues are highlighted by blue labels. **e** Fluorescence signal measurement of the cells transfected by control (mock virus), WT, and N-deletion (residue 85–120 deleted) mKCC2 virus. Transport activity was initiated by the addition of Tl$^+$ (arrow). The inset western-blot shows an expression of WT and N-(85–120) deleted mKCC2. **f** Relative Tl$^+$ transport rate in control ($n = 17$), WT ($n = 14$), and N-deletion ($n = 12$) mKCC2. Unpaired Student's $t$-test was used to evaluate the significance between WT and N-deletion mKCC2 (*$P < 0.05$), and between WT and control (***$P < 0.001$); mean ± SEM.

linking these residues at the intracellular end of the pore to conformational change into the outward-open state that binds bumetanide[29].

mKCC2 exhibits a translocation pathway analogous to hKCC1[39] and hNKCC1 (Fig. 3f, g). At the extracellular end, E224 from TM3 interacts with R142 from TM1b to prevent access from the extracellular side (Fig. 3f) congruent with the apparent gate in hKCC1[39] and above in hNKCC1 (Fig. 3a). At the extracellular end, the narrowest point of mKCC2 is constricted by R142 from TM1b, E224 from TM3, and L577 from TM10 (Fig. 3f). The homologous extracellular gate in KCC4 has also been suggested in previous study[42] in which R140E mutant in mKCC4 significantly reduced the transport activity. Furthermore, analogous to the intracellular essential residues of hNKCC1, R193 from ICL1 of mKCC2 is close to D539 in TM8, forming a polar interaction (Fig. 3g), and from sequence conservation, it appears that all the CCC transporters utilize this interaction (Supplementary Fig. 5).

**Cation-binding sites in hNKCC1.** To probe potential ion-binding sites and gain insight into the mechanism of the translocation pathway, we performed all-atom MD simulations of hNKCC1 and mKCC2 (Supplementary Figs. 10 to 13). In our simulations, hNKCC1 displays several binding sites for Na$^+$ and K$^+$ within the pore region, with distinct preferences for one or the other cation (Supplementary Fig. 11a, b). The most populated Na$^+$ binding site is defined by backbone carbonyls including

L297, W300 in TM1, A610 in TM8, and hydroxyl oxygen atoms of S613 and S614 in TM8 (Fig. 4a)—this site (S1) accounts for 40.6% of TM domain Na$^+$ binding sites in the simulation data (Supplementary Fig. 11a). This proposed site agrees well with the Na$^+$ binding site previously reported in DrNKCC1[40], and the site is also conserved across many Na$^+$ coupled transporters with the LeuT fold[37,47,58–60]. The critical importance of this site is demonstrated by our finding that mutation of S613 or S614 resulted in bumetanide-sensitive $^{86}$Rb$^+$ influx less than 10% of wild-type activity (Fig. 4b and Supplementary Data 1). We also observed that a K$^+$ ion could also access the predominant Na$^+$ binding site S1, albeit with a smaller occupancy probability among all possible K$^+$ binding sites (33.4% of all K$^+$ binding) (Supplementary Fig. 11a, b).

A second proposed cation-binding site (S2) is defined by the backbone carbonyls of P496 and the side chain hydroxyl group of T499 (Fig. 4c). While this cation-binding site binds Na$^+$ with low probability (12.9% of all observed Na$^+$ binding) (Supplementary Fig. 11a), it has a higher probability of occupancy for K$^+$ (33.63%) (Supplementary Fig. 11b). Due to its larger ionic radius, the K$^+$ ion at this site is coordinated by backbone carbonyls of N298, I299 in the helical break of TM1 and hydroxyl oxygen of Y383 (Fig. 4c). In concurrence with the identification of an ion at this position in DrNKCC1[40], and the observation that the tyrosine is conserved in NKCCs and KCCs but not in NCC (which does not transport K$^+$) (Supplementary Fig. 5), we conclude that this is the primary K$^+$ binding site in NKCC1. The crucial role of the Y383 hydroxyl in ion translocation was

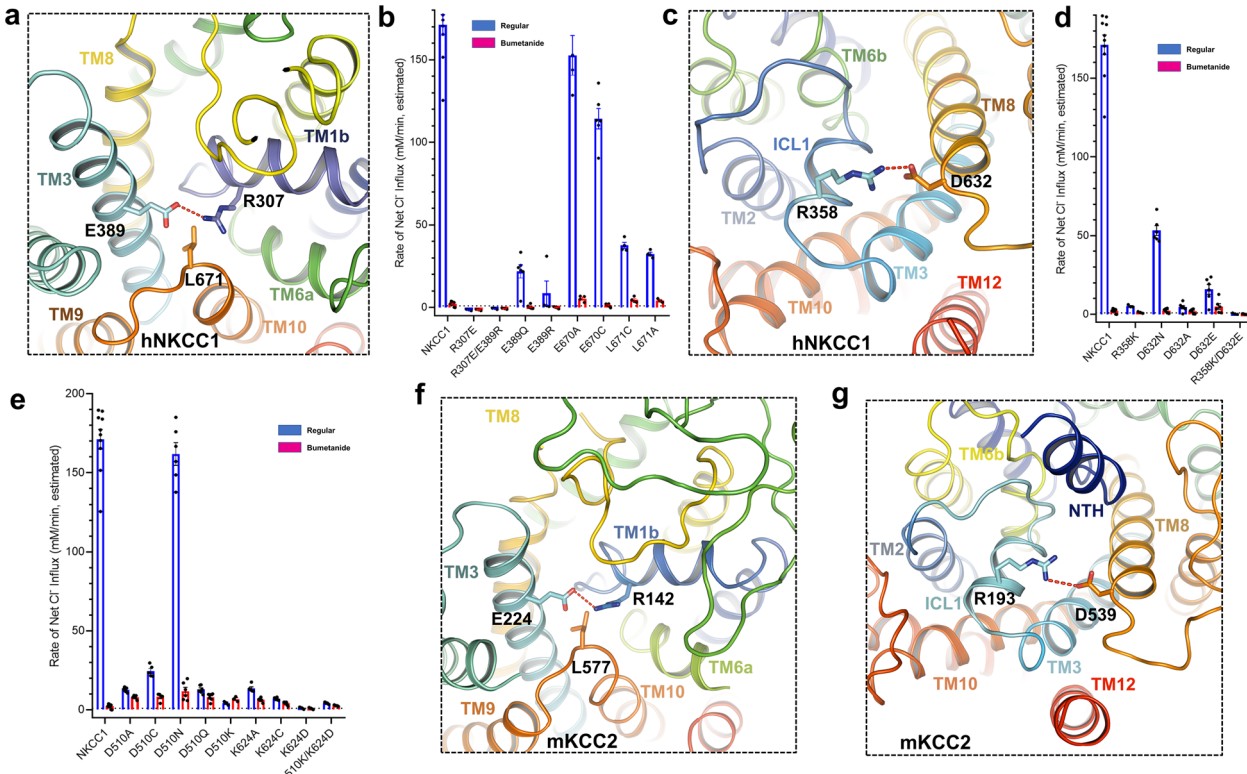

**Fig. 3 Potential gating residues in hNKCC1 and mKCC2. a** The amino group of R307 of TM1b interacts with the carbonyl oxygen of E389 from TM3 to constrict the extracellular gate of hNKCC1. **b** hNKCC1 transport activity is greatly decreased by mutation of the extracellular gating amino acids E389, R307, and L671; mean ± SEM. **c** At the intracellular end of the hNKCC1 translocation pore, R358 of ICL1 interacts with D632 of the neighboring TM8. **d** hNKCC1 transport activity between WT and mutants in the intracellular essential amino acids (R358 and D632); mean ± SEM. **e** hNKCC1 mutants of D510 and K624 at the intracellular end of the pore have greatly reduced transport activity—these residues are implicated in MD simulations (see also Supplementary Fig. 10a, b); mean ± SEM. **f** Homologous to hNKCC1 (**a**), the extracellular gate of mKCC2 is restricted by hydrogen bonds between R142 of TM1b and E224 of TM3. **g** Potential intracellular-gating residues of mKCC2: homologous to hNKCC1 (**c**), R193 of ICL1 in mKCC2 is close to D539 of TM8.

demonstrated by the previous finding that Y383C, Y383W, and Y383F mutations exhibit impaired transport activity[29]. Indeed, in our MD simulations, we found that the hydroxyl group of Y383 not only directly coordinates the K+ ion via the hydroxyl oxygen, but also attracts a Cl− ion via hydroxyl O–H···anion hydrogen bond (Supplementary Fig. 10d).

As noted above, the interaction between E389 of TM3 and R307 of TM1b blocks the pore from the outside and thus appears to form the extracellular gate (Fig. 3a). We have noted in some of our well-tempered metadynamics simulations that the salt bridge interaction at this site is replaced by Na+ or K+ binding, thus opening the gate (Supplementary Fig. 10e), and we also saw that cations could be bound by the cluster of acidic residues including E389, E393, E405, and E670 (S3 binding site in Supplementary Fig. 11a, b). These results, therefore, suggest that in the reverse mode of the overall transport mechanism the transition from the inward-open to outward-open conformations may be triggered by cations breaking the salt bridge at E389-R307.

**Anion binding sites in hNKCC1.** Cl− ion movement is frequently accompanied by the movement of cations in our MD simulations, indicating that Cl− transport is facilitated by Na+ and K+. The first potential Cl− binding site 1 (Cl− 1 site) is a passive binding site due to the presence of K+ ion at its primary binding site, which contributes 4.1% occupancy probability over all possible Cl− binding sites (Supplementary Fig. 11c), defined by backbone carbonyls of G301, V302, and M303 (Fig. 4c), akin to

the site1 Cl− binding site in DrNKCC1[40]. A second proposed Cl− binding site (Cl− 2 site) accounts for 17.6% occupancy probability and is located at the middle of the TM domain, defined by the backbone carbonyls of G500, I501, L502, and the hydroxyl group of Y686 (Fig. 4d and Supplementary Fig. 11c)—this site was also identified as a Cl− 2 site by MD and cryo-EM density in DrNKCC1[40], showing strong conservation and supporting the suggestion that this is the binding site for one of the two transported Cl− ions[40]. We also found a dominant Cl− binding region (Cl− 3 site) at the intracellular end of the open pore, contributed by the side chains of R294, N298, and N506 (Supplementary Figs. 10f and 11c). This site accounts for 73.5% of Cl− binding identified in hNKCC1 (Supplementary Fig. 11c) and is congruent with the transient binding site of DrNKCC1[40] and a site in the CLC transporter[61].

To explore the co-occupancy of bound ions in hNKCC1, we further performed 9 ×1000 ns regular MD simulations containing all four ions in the TM domain (Supplementary Fig. 12). The primary ion-binding sites in the TM domain of hNKCC1 were obtained based on multiple walkers well-tempered metadynamics simulations, and these are consistent with the proposed ion-binding sites of DrNKCC1[40]. A representative MD trajectory (Fig. 4e) demonstrates the stability of four ions in their respective binding sites during 1000 ns simulations in hNKCC1, further confirming the conservation of ion-binding properties in the NKCCs.

TMs 1, 6, and 8 are seen to contribute to the majority of binding site residues responsible for the coordination of ions in the transport process. To begin to examine this by systematic

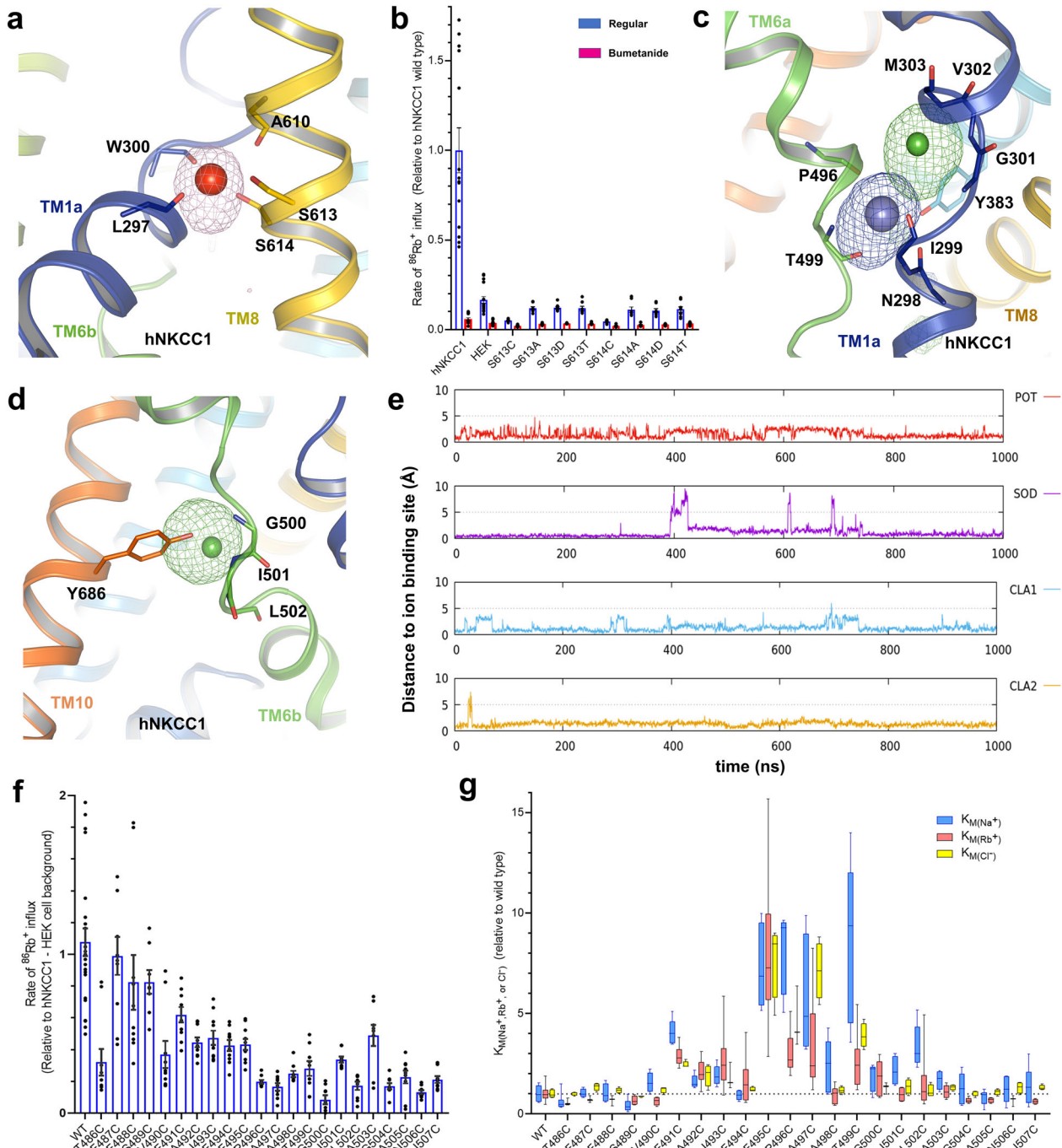

**Fig. 4 Ion-binding probability densities and mutagenesis assay of hNKCC1. a** The $Na^+$ binding site of hNKCC1, based on molecular dynamic simulations. $Na^+$ is stabilized by interactions with S613, S614, and A610 in TM8, and L297 and W300 in TM1. **b** hNKCC1 transport activity is greatly decreased by mutations of S613 and S614 in the proposed $Na^+$-binding site; mean ± SEM. **c** $K^+$ and $Cl^-$ ($Cl^-$-1) binding sites of hNKCC1 based on molecular dynamic simulations. The $K^+$ ion is stabilized by interaction with Y383 from TM3, N298, and I299 in TM1, P496, and T499 in TM6 and by the $Cl^-$ ion. The $Cl^-$ ion is stabilized by interaction with G301, V302, and M303 from TM1 and by the $K^+$ ion. **d** A second $Cl^-$ binding site in hNKCC1 ($Cl^-$-2), based on molecular dynamic simulations. This $Cl^-$ ion is stabilized by interactions with G500, I501, and L502 in TM6a, and Y686 in TM10. **e** Time evolution of the distance of the four ions ($Na^+$, $K^+$, $Cl^-$-1, and $Cl^-$-2) from its proposed binding sites based on molecular dynamics simulation. **f** hNKCC1 $^{86}Rb^+$ transport activity of cysteine mutants in a scan of TM6; mean ± SEM. **g** The relative $K_M$ (for $Na^+$, $Rb^+$, and $Cl^-$) for activation of transport of each of the cysteine mutants in TM6 as a ratio to the wild-type hNKCC1. The absolute $K_M$s of wild-type hNKCC1 were 5.6, 3.3, and 18.3 mM for $Na^+$, $Rb^+$, and $Cl^-$, respectively. "Whiskers" indicate the maximum range of the values.

mutagenesis, we carried out a cysteine scan of TM6 and examined the effect of individual mutations on $^{86}Rb^+$ transport. As illustrated in Fig. 4f, mutations to cysteine throughout this TM domain resulted in a loss of maximal transport with most changes in the intracellular half resulting in less than 30% of activity; this includes P496, T499, G500, and L502 identified above as being directly involved in $K^+$ or $Cl^-$ binding. G500C, in particular, is almost completely inactive, likely reflecting the need for a flexible glycine at the end of the "un-wound" region separating TM6a and 6b.

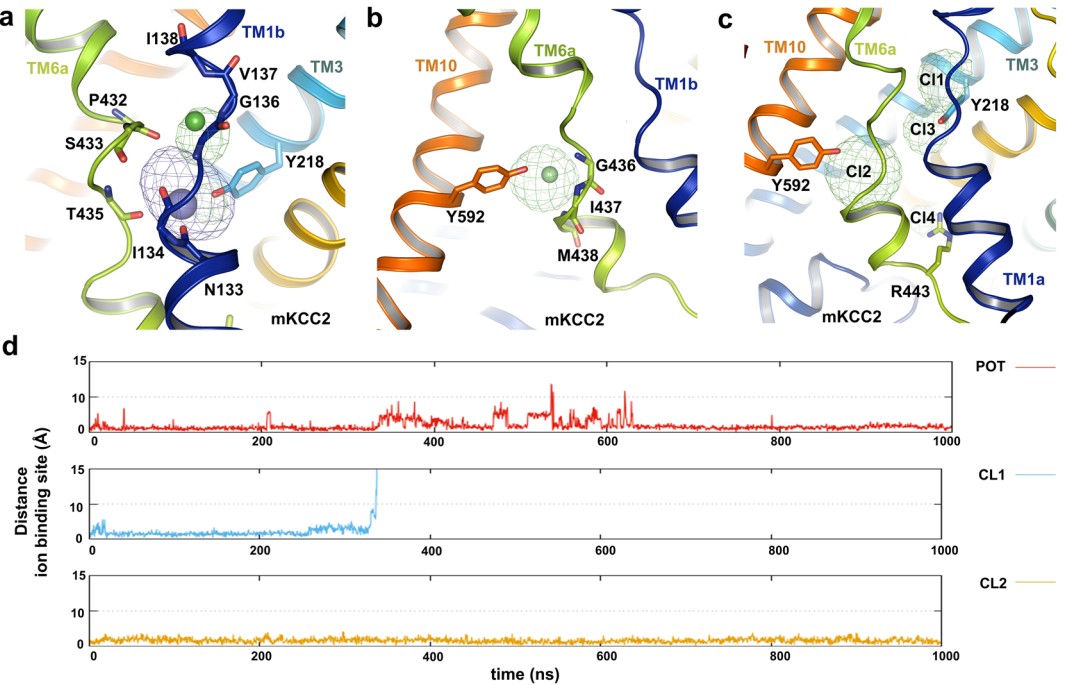

**Fig. 5 Ion-binding probability densities of mKCC2. a** K$^+$ and Cl$^-$ (Cl$^-$-1) binding sites of mKCC2 based on molecular dynamic simulations. Similar to hNKCC1, K$^+$ is stabilized by the interaction with Y218 in TM3, N133 and I134 in TM1a, T435 and P432 in TM6, and the Cl$^-$ ion. The Cl$^-$ binding site is stabilized by interaction with G136, V137, and I138 in TM1 and by the K$^+$ ion. **b** A second Cl$^-$ binding site (Cl$^-$-2) of mKCC2. Similar to (Cl$^-$-2) in hNKCC1, this Cl$^-$ is stabilized by the interaction with G436, I437, and M438 in TM6, and Y592 in TM10. **c** Four Cl$^-$ ion probability densities in mKCC2 based on molecular dynamic simulations. **d** Time evolution of the distance of the three ions (K$^+$, Cl$^-$-1, and Cl$^-$-2) from its proposed binding sites based on molecular dynamics simulation. The Cl$^-$ ion escapes from the Cl$^-$-1 site to the neighboring Cl$^-$-3 site after 300 ns.

We also examined the apparent ion affinity of the TM6 cysteine mutants for each of the three transported ions. In addition to the changes to maximal activity, mutations to residues in the ion-binding region (F495–T499) produced a 4–8-fold decrease in the apparent affinity for ion binding (Fig. 4g and Supplementary Data 1). The largest effect was on K$_M$ for Na$^+$— quite surprising since residues P496 and T499 appear to be directly involved in K$^+$ binding. While some degree of kinetic coupling in the ordered transport model is to be expected[28], kinetics alone cannot explain this result. We suggest the answer may lie in the flexibility of TMs 1 and 6 and the fact that TM1 participates in both ion-binding sites, so that induced fit at one site could affect the disposition of TM1 and thus indirectly affect the other site. This explanation may also be extended to explain the strictly-ordered binding of Na$^+$ and K$^+$ in ion translocation[10,11] where no such ordering is obvious in the location of the respective ion-binding sites. Further evaluation should probably await the availability of an outward-open structure since the apparent affinities measured in the experiments of Fig. 4f, g are related to the binding of ions from the extracellular medium.

**Ion-binding sites in mKCC2.** We performed similar MD simulations with mKCC2 to evaluate the potential K$^+$ binding site (Supplementary Figs. 10, 11, and 13). We found that the highest probability K$^+$ binding (55% of all K$^+$ binding) is at a site defined by the backbone carbonyls of N133, I134, P432, S433, T435, and the hydroxyl group of Y218 (Fig. 5a and Supplementary Fig. 11d), equivalent to the K$^+$ binding site identified by cryo-EM and molecular dynamics in hNKCC1[39] and mKCC4[42]. Our simulations also found that K$^+$ binds with lower frequency (18% for all K$^+$ binding) at the intracellular entrance of the pore, at a site made up by the backbone carbonyls of P129, A439, and

the side chain of N442 (Supplementary Figs. 10g and 11d). Finally, we noted a small fractional occupancy (7.8% for all K$^+$ binding) on the extracellular side around E224 (Fig. 3f and Supplementary Fig. 11d) similar to extracellular cation-binding noted in hNKCC1 (Supplementary Fig. 11a, b).

Our simulations identify the principal Cl$^-$-binding site (31.85%, Cl$^-$ 1 site) in the middle of the TM domain associated with the bound K$^+$ ion (Fig. 5a, c and Supplementary Fig. 11e) and defined by main-chain amide groups of G136, V137, and I138 (Fig. 5a), analogous to the Cl$^-$ 1 site identified by cryo-EM in hNKCC1[39]. Similar to Y383 in hNKCC1, Y218 in mKCC2 also has a bifunctional role by stabilizing both cation and anion (Fig. 5a, c). We also identified a second Cl$^-$ ion-binding site (with 22% occupancy, Cl$^-$ 2 site) (Supplementary Fig. 11e) analogous to the Cl$^-$ 2 site in hNKCC1[39] and mKCC4[42] and corresponding to the Cl$^-$ 2 site in hNKCC1—this site is made up of a backbone of G436, I437, M438, and the hydroxyl group of Y592 (Fig. 5b). We performed MD simulations to explore the co-occupancy of bound ions in mKCC2 (Supplementary Fig. 13). In the case of mKCC2 with a bound K$^+$ ion, Cl$^-$ does not appear stable at the Cl$^-$ 1 site, but because of interaction with the Y218 hydroxyl, the Cl$^-$ ion escapes to a neighboring Cl$^-$ 3 site after ~300 ns of our regular MD simulation (Fig. 5d and Supplementary Fig. 13d).

**Water in hNKCC1.** Our structures and molecular simulations can also provide insight into the possibility of water transport through CCCs. Many transporters are thought to provide transient pathways through which water can move, an idea that has been well-supported by MD simulations[62,63]. More controversial is the hypothesis that some transporters can function as "water-pumps" as supported by physiological evidence that NKCC1 mediates the active transport of 600 water molecules in one turnover event, coupled to the movement of the 4 cotransported

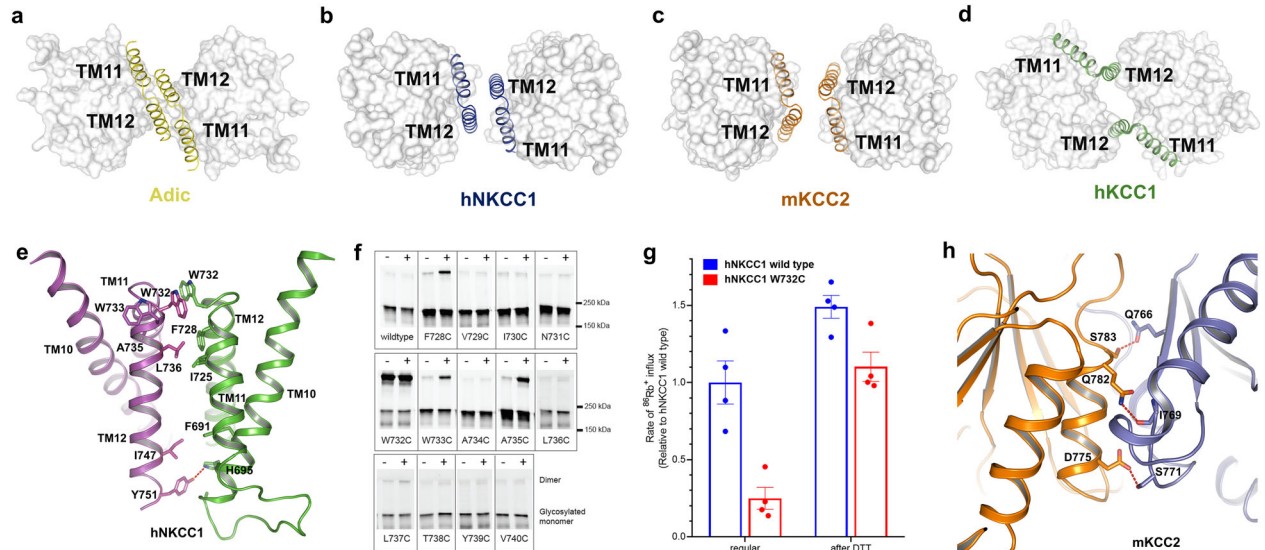

**Fig. 6 Dimeric interface of hNKCC1 and mKCC2. a–d** Schematic representation of the TMD dimer architecture between different transporters (Adic (yellow), hNKCC1 (blue), mKCC2 (brown), and hKCC1 (green)) with TM11-12 shown as ribbons and other TMD regions shown as surface representation. **e** The NKCC1 homodimer is stabilized by hydrogen bond interaction between Y751 in TM12 of one monomer with H695 in TM10 of the other; in addition, there are stabilizing hydrophobic interactions between TM11-12 hairpins (labeled residues). **f** Homodimeric oxidative crosslinking of NKCC1 with single cysteine substitutions in TM 11-12. Cells were treated with (+) or without (−) cupric phenanthroline before lysis. **g** The W732C mutant exhibits about 25% of the Rb+ influx activity of wild-type NKCC1. Transport activity is largely restored in W732C after a 30 min pretreatment with 40 mM DTT to break the disulfide linkage between hNKCC1 monomers; mean ± SEM. **h** Cytoplasmic-domain dimeric interactions in mKCC2. D775, Q782, and S783 from one subunit (brown) form hydrogen bonds with Q766, I769, and S771 from the neighboring subunit (blue) to stabilize the dimer.

ions[64]. In Supplementary Fig. 14, we examine water in hNKCC1 during one of our simulations of the inward-open transporter— there is indeed a continuous pathway that is occupied by water molecules extending through the transporter pore (Supplementary Fig. 14a, b and Supplementary Data 2), strongly supporting the idea that water can move passively through NKCC1 in this conformational state.

While the possibility of water permeation through NKCC1 is clear from these data, the actual number of water molecules in the interior of the transporter is relatively small (Supplementary Fig. 14c). There are only about 40 water molecules in the narrow central pore between 8 and −12 Å on the Z axis, and including the vestibules at intracellular and extracellular ends there are only about 86 water molecules between 12 and −14 Å. This is very far from the 600 molecules in the water-pump hypothesis[64], and we believe that similar to the conclusion for the $Na^+$-glucose transporter[62], these findings rule out the idea of NKCC1 as an isosmotic water pump.

**The architecture of NKCC and KCC dimers**. As illustrated in Fig. 6a–d, the basic TM-dimer architecture of APCs has been conserved from bacterial transporters to vertebrates—in this dimer motif the TM11-12 helical hairpins from the two halves of the dimers are apposed with one another, rotated 180° in a slightly offset arrangement. This aspect of the structure is remarkably similar for the bacterial amino acid transporter AdiC[34] and eukaryotic NKCCs[40] (Fig. 6a, b). mKCC2 retains the same basic TM11-12 architecture (Fig. 6c), although the TM11-12 hairpins tilt away from one another towards the intracellular end. This tilt at the KCC2 interface results in a significantly larger gap between the TM domains of mKCC2 compared to NKCC1— consequently, less of the overall dimer interaction is contributed by TM interactions in mKCC2 (Fig. 1a, d).

Anticipating the potential for homodimeric crosslinking at the center of symmetry between the two monomers we used a cysteine scanning and crosslinking strategy to examine the TM11-12 dimer interface in hNKCC1 (Fig. 6e). We scanned 20 residues in TM11-12, mutating to cysteine; each was properly expressed at the plasma membrane. One of these (W732C) was found to form a spontaneously crosslinked homodimer pinpointing W732 as a residue of native contact (Fig. 6f). This crosslink was found to be inhibitory, and the inhibition was largely relieved by disulfide reduction with DTT treatment (Fig. 6g and Supplementary Data 1), indicating a functional role for the dimer interaction. Using cupric phenanthroline to promote disulfide formation between cysteines in close proximity to one another, we found additional crosslinking with mutants F728C, W733C, and A735C, showing that these residues also reach homodimeric proximity, albeit with lower frequency (Fig. 6e, f). Nearly identical results were obtained in the context of a C723S-C724V double mutation that removes two potentially interfering cysteines in TM11 and a similar pattern of crosslinking was obtained with iodine and with bifunctional MTS reagents (Supplementary Fig. 15). These results are consistent with the placement of F728C, W732C, W733C, and A735C on the dimer interface of the TM11-12 hairpin (Fig. 6e).

The dimer architecture in the structure of hKCC1[39] stands in striking contrast to the well-conserved dimer motif observed in AdiC and in the inactive state of eukaryotic transporters DrNKCC1, hNKCC1, and mKCC2—the dimer interface of hKCC1[39] is in fact displaced by 15.6 Å and rotated 31° relative to mKCC2 (Fig. 6c, d and Supplementary Fig. 3e). It thus seems likely that the hKCC1 structure represents a different functional state of the KCC transporter, quite likely an active (dephosphorylated KCC) state. It is also notable that while TM1-11 of the KCC1 and KCC2 monomers are virtually superimposable, TM12 is seen to undergo a relative tilt of 23° in KCC1 (Fig. 1f). A relationship between activation of CCCs and dimer interaction and TM12 has been suggested in FRET[22] and crosslinking studies[23] and will be further discussed below.

Along with this large difference in TM dimerization between hKCC1 and mKCC2 structures, there appear to be major

differences in the C termini. Our structure of mKCC2 includes the C-terminal intracellular region with the C terminus of each subunit connected to the TM domain by an alpha helix (scissor helix[30]) and wrapping around to the neighboring subunit as in NKCC1 (Fig. 1d and Supplementary Fig. 3c). Just as in the reported cyanobacterial MaCCC domain[65], the *C. elegans* CCC domain[66], and in DrNKCC1[40], each mKCC2 CTD includes 10 β sheets and 8 bridging α helices (Supplementary Fig. 16). The domain-swap arrangement of C termini appears well stabilized by the respective scissors helices and by a CTD dimer interface consisting of three hydrogen bonds between helices α3 and strands β3 as in DrNKCC1[40] (D775-S771, Q782-I769, and S783-Q766, Fig. 6h). On the other hand, the location of the hKCC1 C termini appears to be too poorly fixed relative to the TMs to have permitted structural assignment of the hKCC1 C termini, even though the TM domain was resolved to 2.9 Å resolution[39]. Thus, from the two KCC structures, it appears that the conformational change involved in TM dimer relationships is also associated with freeing of the C termini from the condensed domain-swap configuration.

Interestingly, although DrNKCC1 and mKCC2 structures share the basic feature of domain-swapped C termini and are both presumed to represent inactive states, they differ in important details (Supplementary Fig. 17). As illustrated in Supplementary Fig. 17d, when aligned by the TM domain, the C termini of mKCC2 are rotated ~85° relative to DrNKCC1. It seems likely that the condensed domain-swap arrangement of the CTD is central to assuring a stable deactivated state in both transporters but that other mechanistic details are different in order to provide an activating mechanism with N-terminus phosphorylation in NKCC but with C-terminus dephosphorylation in KCC.

## Discussion

The structures of CCCs that are now available provide a great deal of insight into mechanisms by which the Na(K)CCs and KCCs may be regulated. As illustrated by the cartoon models in Fig. 7, we propose that the inactive state of these transporters is characterized by a TM-domain dimer interface at apposed TM11-12 hairpins, and with CTD domains tightly organized beneath the TMs in a domain-swap conformation (Fig. 7a). Our evidence is that both available NKCC structures, as well as our mKCC2 structure, are in this inactive configuration. On the other hand, we propose that the hKCC1 structure represents an active state which is marked by the absence of an N-terminus inhibitory peptide at the intracellular end of the pore, and by displacement of the TM-domain interface, tilting of TM12, and the freeing of the CTDs into a tethered arrangement. Support for this activation-inactivation model has been provided as a decrease in FRET between NKCC1 dimer CTDs upon activation[22], and by a change in the TM10-12 architecture as found in crosslinking studies[23]; some caveats are provided in Supplementary Note 1.

The exact mechanisms of activation must differ between NKCCs and KCCs since phosphorylation in the N terminus activates the former and phosphorylation in the CTD inactivates the latter. For NKCCs, we propose that the non-phosphorylated N terminus does not significantly interact with the rest of the protein and that the domain-swapped CTD has an inhibitory interaction with the TM-domain. We further propose that upon phosphorylation the N terminus interacts with the CTD to disrupt the condensed structure and promote the active CTD-tethered conformation (Fig. 7b).

A key feature in the inactive-state structure of mKCC2 is the presence of a strategically placed region of the N terminus, which blocks the intracellular end of the translocation pore and must

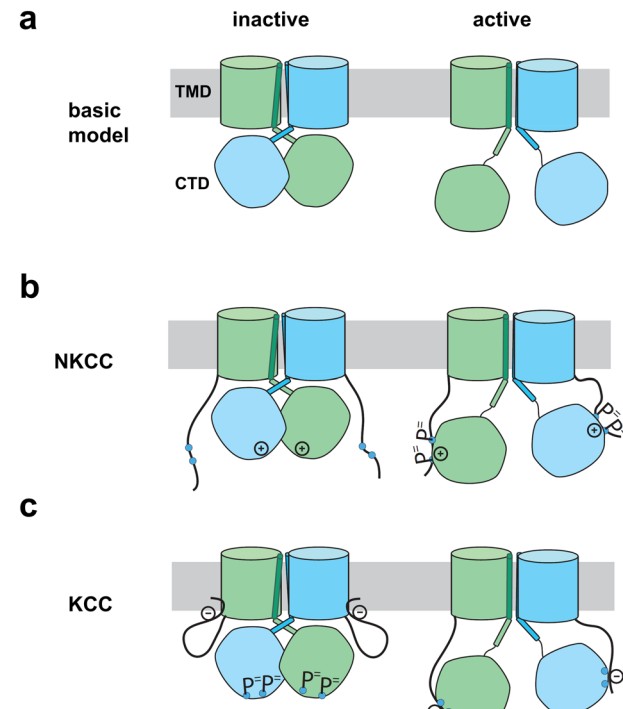

**Fig. 7 Hypotheses of regulatory activation in CCCs. a** The basic model of cation-chloride cotransporter activation. It is proposed that the inactive state is characterized by an inward-open TM conformation with CTDs in a tightly organized domain-swap configuration. On activation, the CTDs are released to a freely tethered configuration with accompanying change in the dimer architecture. **b** Our hypothesis is that NKCC activation occurs when the phosphorylated N terminus interacts with the CTD and disrupts the dimer. **c** The KCC inactive state exhibits an N terminus inhibitory peptide blocking the transport pore. Activation occurs when the peptide instead binds to the non-phosphorylated C terminus.

therefore inhibit transport (Fig. 2). The presence of this inhibitory peptide suggests that CTD-TM interactions may be less important in the inactivation of KCCs compared to their role in NKCCs, which is quite possible in light of the different positions of the CTD relative to the TM-domain (Supplementary Fig. 17d). The KCC N terminus includes two regions of negative charge including 13 acidic residues, present in all four KCC isoforms (Supplementary Fig. 5). We propose that this is a key element in the activation mechanism, which acts to prevent interaction of N and C termini when the C terminus is phosphorylated, but allows interaction with the non-phosphorylated CTD and thus promotes the loosely tethered active conformation (Fig. 7c).

The results provided here identify probable gating residues and ion-binding sites in hNKCC1 and mKCC2, and we present functional studies that strongly support these hypotheses—our results are fully congruent with assignments that have been made in DrNKCC1[40], mouse KCC4[42], and hKCC1[39]. All the structures obtained to date are for the inward-open conformation, with binding sites exposed to the intracellular medium. We can infer a transport conformational change and the nature of extracellularly exposed sites from homology modeling[29] with the occluded and outward-open structures of distantly related APC superfamily members AgcS[37], AdiC[34,45], and ApcT[35], but a complete understanding of the transport cycle awaits the availability of occluded and outward-open structures of NKCC and KCC. It will be especially instructive to see how structure determines the strictly-ordered (Na$^+$, Cl$^-$—K$^+$, Cl$^-$) extracellular binding that is predicted by transport studies[10,11].

Recent evidence has tied NKCC1 and KCC2 dysfunction to many neurological diseases of which human epilepsy is the most common worldwide. Epilepsy is a disease of abnormal neuronal network synchronization, in many cases due to decreased KCC2 expression or activity and lowered $[Cl^-]_i$[67–71]; KCC2 has also been implicated as an important factor in neuronal morphogenesis, particularly in the development of dendritic spines[68,72]. As illustrated in Supplementary Fig. 18, the mKCC2 structure provides a framework to understand the mechanisms of pathogenesis in epilepsy-related KCC2 mutations, of which fifteen have recently been identified[73,74]. Biochemical studies have characterized a number of these as having decreased KCC2 expression —this likely accounts for at least part of a defect in $Cl^-$ balance in L288H, S376L, L403P, G528D, and R952H[67–69,75] (we refer to human mutations in hKCC2b numbering). Three mutations are in TM helices implicated in ion binding (A191V in TM3, L403P, and M415V in TM6), including M415V identified as part of the $Cl^-$ 2 site (M438 in mKCC2, Fig. 5b)—we expect that these mutations directly affect the translocation process. Finally, many of the epilepsy-linked mutations appear likely to involve regulatory defects, either in phosphorylation or inter-domain or inter-subunit regulatory interactions or effects on morphogenesis —these mutations include R952H and R1049C affecting phosphorylation[67], E50-Q93 del in the N terminus, S748 del located in the C-terminus dimer interface, G528D in ICL4 potentially interacting with either N- or C-terminus residues, and L288H and S376L close to the ECD dimer interface. With early efforts directed to increasing intrinsic KCC2 activity and membrane trafficking showing some promise[76,77], it is anticipated that further understanding of the KCC2 mechanisms will lead to novel development of therapeutic approaches to associated neurological disorders.

The structures of hNKCC1 and mKCC2, considered with other resolved CCC proteins, provide a critical window into the mechanism of ion translocation by the cation-chloride cotransporters and the mechanism by which the transporters are regulated by the WNK-SPAK/OSR1 kinase pathway. These structures have allowed us to propose a set of hypotheses for the regulation of the cation-chloride cotransporters, which will be tested in a broad range of future experiments. The understanding gained from these efforts will guide investigations of NKCCs and KCCs as therapeutic targets in disorders of ion transport and in a broad spectrum of neurological diseases.

## Methods

**Cell culture and transfection**. SF9 cells (ATCC CRL-1711) were cultured in Sf-900 III SFM (ThermoFisher Scientific) supplemented with 1 × penicillin/streptomycin (Solarbio) at 27 °C. HEK293F cells were cultured in SMM 293-TI medium (Sino Biological Inc.) supplemented with 1 × penicillin/streptomycin (Solarbio) at 37 °C with 8% $CO_2$. DNA for the *Homo sapiens* NKCC1 and *Mus musculus* KCC2 were obtained from GenScript and cloned into the PEG BacMam vector for virus amplification and protein purification. A twin-Strep affinity tag was inserted after the C terminus of NKCC1, and the same tag was inserted in front of the N terminus of KCC2. Baculovirus of both NKCC1 and KCC2 were generated in sf9 insect cells under the Baculovirus expression protocol of ThermoFisher Scientific. HEK293F cells were transfected using the P3 BacMam virus with a volume ratio of 1:30 (virus: HEK293F cell) when cell density reached $2.5 \times 10^6$ cells/ml. Twelve hours after transfection, 10 mM sodium butyrate was added to the cell culture and cells were cultured for 48 h before harvesting.

Crosslinking studies and functional assays were carried out with full-length hNKCC1 tagged with flag and YFP, encoded by a synthetic cDNA, and expressed in HEK293 cells—methods for cassette mutagenesis, transfection, and growth of cells have been previously described in detail[23,78].

**Protein expression and purification**. For each batch of hNKCC1 and mKCC2 protein purification, two liters of transfected HEK293F cells were harvested by centrifugation at 3000×*g*. Cell pellets were resuspended in lysis buffer containing 20 mM Hepes, pH 7.4, and 150 mM NaCl, 1 μg/ml leupeptin, 1.5 μg/ml pepstatin, 0.84 μg/ml aprotinin, 0.3 mM PMSF and lysed by sonication for 5 min. The cell membrane was pelleted after a 100,000×*g* ultracentrifugation for 1 hour. The

membrane was resuspended in buffer containing 20 mM Hepes, pH 7.4, 150 mM NaCl, 2 mM DTT, and 1% (w/v) digitonin for 2 h with gentle rotation at 4 °C. After ultracentrifugation at 100,000×*g* for 20 min, the supernatant was incubated with Strep-Tactin Sepharose (IBA) for 1 h with gentle rotation at 4 °C. The resin was washed extensively with wash buffer containing 20 mM Hepes, pH 7.4, 150 mM NaCl, 2 mM DTT, and 0.1% (w/v) digitonin. The target hNKCC1 and mKCC2 proteins were eluted with wash buffer plus 5 mM D-Desthiobiotin (IBA) and concentrated to a final volume of approximately 100 μl. The final proteins were applied to size-exclusion chromatography (Superpose-6 10/300 GL, GE Healthcare) in buffer containing 20 mM HEPES (pH 7.4, 150 mM NaCl, and 0.1% digitonin). The peaks corresponding to hNKCC1 and mKCC2 proteins were collected for further cryo-microscopy analysis (Supplementary Fig. 1). Unprocessed original blot images for the protein expression are provided in Supplementary Fig. 19.

**Cryo-electron microscopy**. The cryo-EM grids were prepared using a Vitrobot Mark IV (FEI) operated at 8 °C and 100% humidity. For samples of hNKCC1 and mKCC2, 4 μl aliquots of fresh samples at concentrations of ~10 mg/ml were applied onto glow-discharged holey carbon grids, 300 mesh gold (Quantifoil R1.2/1.3). After 5 s, the grids were blotted for 2 s and plunged into liquid ethane for quick freezing.

The cryo-EM grids were screened on a Tecnai Arctica microscope (FEI) operated at 200 kV using a Falcon II 4k × 4k camera (FEI). Qualified grids were transferred to a Titan Krios microscope (FEI) operated at 300 kV for data acquisition and equipped with Cs-corrector (ThermoFisher Scientific Inc.), Gatan K2 Summit detector, and GIF Quantum energy filter. Images were automatically recorded using AutoEMation with a slit width of 20 eV for the energy filter and in super-resolution mode at a nominal magnification of 105,000×, corresponding to a calibrated pixel size of 1.091 Å at object scale, and with defocus ranging from 1.4 to 1.9 μm. Each stack was exposed for a total of 5.6 s with an exposure of 0.175 s for each of 32 frames—the total dose for each stack was about 50 $e^-/Å^2$.

**Image processing**. Simplified flowcharts for data processing of hNKCC1 and KCC2 are summarized in Supplementary Fig. 1. In total, 2082 and 1642 micrographs (movie stacks) were collected for hNKCC1 and mKCC2 samples. Motion correction was performed using MotionCor2[79], generating summed micrographs with or without dose weighting. CTFFIND4[80] was used to estimate the contrast transfer function (CTF) parameters and produce the CTF power spectrum based on summed micrographs from MotionCor2. Particles were auto-picked on summed micrographs from MotionCor2 using RELION-3[81]. For both hNKCC1 and mKCC2, ~1000 particles were manually picked in advance, and processed by 2D classification using RELION-3. The resulting 2D averages served as the templates for particle picking.

For the hNKCC1 data set, 622k particles were auto-picked from 2082 micrographs and two rounds of 2D classifications were performed to exclude noise and other bad particles. 561k particles from qualified 2D averages were selected for further 3D analysis. Ahead of 3D classification, a round of refinement was applied on the whole particle sets using RELION-3. Three rounds of 3D classification with C1 symmetry generated 80k particles with a good signal. Each particle was re-centered using the in-plane translations measured in 3D refinement and re-extracted from the motion-corrected integrated micrographs. Gctf was used to refine the local defocus parameters[82]. The final particles were processed by auto-refine with soft mask and C2 symmetry imposed using RELION-3, resulting in a 3.5 Å resolution map of hNKCC1 with C2 symmetry (Supplementary Fig. 1g). However, the final 3.5 Å map of hNKCC1 only comprises the transmembrane regions, as the cytoplasmic regions are unclear. In addition to this high-resolution map of hNKCC1, we also obtained a medium resolution map (6.2 Å) of hNKCC1 including cytoplasmic regions, after several rounds of 3D classification and refinement.

For the mKCC2 data set, the procedure was similar to hNKCC1. After extensive rounds of 2D classification, 318 k particles were applied to 3D classifications; two rounds of 3D classification with C1 symmetry generated 66 k particles, and after auto-refine with soft mask and C2 symmetry, we obtained a 3.8 Å resolution map of mKCC2 (Supplementary Fig. 1h).

**Model building**. Before model building, models of full-length hNKCC1 and mKCC2 were predicted on I-TASSER and Phyre2 online servers[83]. Sequence alignment and secondary structure prediction of hNKCC1 and mKCC2 were used to aid the model building. The predicted model of hNKCC1 and mKCC2 were docked into the cryo-EM map with a resolution of 3.5 and 3.8 Å in Chimera and manually adjusted in Coot to acquire the atomic model of hNKCC1 and mKCC2, respectively[84,85]. Model refinement was performed on the main chain of the two atomic models using the real_space_refine module of PHENIX[86] with secondary structure and geometry restraints to avoid over-fitting. For cross-validation against over-fitting, we randomly displaced the atom positions of the final model by up to a maximum of 0.5 Å[87] and refined against the half map 1 generated by RELION 3D auto-refine procedure, resulting in a model named Test. We then calculated the FSC curve of both half maps against the model Test and compared it with the FSC curve of the final model against the summed map generated by the RELION 3D auto-refine procedure. All reported resolutions are based on the gold-standard FSC

= 0.143 criteria[88], and the final FSC curves are corrected for the effect of a soft mask using high-resolution noise substitution[89]. Final density maps were sharpened using RELION, and local resolution maps were calculated using ResMap[90]. Models with lipids were subjected to global refinement and minimization in real space refinement using PHENIX[86].

**Sample preparation and mass spectrometry.** Gel bands of hNKCC1 and mKCC2 were excised and digested in-gel, and proteins were identified by mass spectrometry. Briefly, proteins were disulfide reduced with 25 mM dithiothreitol (DTT) and alkylated with 55 mM iodoacetamide, in-gel digestion was performed using sequencing grade-modified trypsin in 50 mM ammonium bicarbonate at 37 °C overnight, peptides were extracted twice with 1% trifluoroacetic acid in 50% acetonitrile aqueous solution for 30 min, and volume was reduced in a SpeedVac.

For LC-MS/MS analysis, peptides were separated by a 60 min gradient elution at a flow rate of 0.300 μl/min with the EASY-nLC 1000 system which was directly interfaced with the Thermo Orbitrap Fusion mass spectrometer. The analytical column was a homemade fused silica capillary column (75 μm ID, 150 mm length; Upchurch, Oak Harbor, WA) packed with C-18 resin (300 A, 5 μm; Varian, Lexington, MA). Mobile phase A consisted of 0.1% formic acid, and mobile phase B consisted of 100% acetonitrile and 0.1% formic acid. The Orbitrap Fusion mass spectrometer was operated in the data-dependent acquisition mode using Xcalibur3.0 software with a single full-scan mass spectrum in the Orbitrap (350–1550 *m/z*, 120,000 resolution) followed by three seconds data-dependent MS/MS scans in an Ion Routing Multipole at 30% normalized collision energy (HCD). The MS/MS spectra from each LC-MS/MS run were searched against the selected database using the Proteome Discovery searching algorithm (version 1.4). The phosphopeptides were further verified using the phosphoRS 3.1 node in Proteome Discoverer software, which determines the localization of phosphorylation sites within validated peptide sequences. All MS/MS spectra corresponding to phosphopeptides were manually examined.

**Thallium ion (Tl$^+$) flux transport assay.** The FluxOR$^{TM}$ II green potassium ion channel assay kit (F20016, ThermoFisher Scientific) was adapted to estimate the KCC2 transporter activity as previously described[56,57] with modifications. Briefly, Sf9 insect cells at a density of $1.5-2 \times 10^6$ cells/ml were transfected by the mock baculovirus, WT mKCC2 baculovirus, and N-deletion mKCC2 (residues 85–120 deleted) baculovirus, respectively. After 36-48 hours of suspension culture, 60 μl of cells were plated to the poly-lysine treated 96-well plate for 3 h to allow sufficient adhesion. Growth media was replaced with 1× loading buffer and incubated at 27 °C from light for 1 hour. Dye-loaded cells were then washed twice with 80 μl assay buffer containing 135 mM NaCl, 1 mM MgCl$_2$, 1 mM Na$_2$SO$_4$, 1 mM CaCl$_2$, 15 mM HEPES, 2.7 mM Probenecid, and 10 μM bumetanide. The fluorescence was measured on a Perkin-Elmer Envision Multilabel Plate Reader (excitation/emission wavelength 490/525 nm). The recording baseline was measured in 80 μl assay buffer by the first 40 s before Tl$^+$ addition. The transport assay was initiated by adding 20 μl of 5 mM thallium sulfate (5× concentrated in assay buffer) and recorded for 150 s. The initial rate of Tl$^+$ transport was deduced by analyzing the linear increase of fluorescent signals within the initial 10 s following the Tl$^+$ addition. When measuring the effect of DIOA on transport activity, DIOA was added to a concentration of 100 μM for at least 10 minutes before the initiation of the transport. Unpaired Student's *t*-test was used to evaluate the significance between WT and N-deletion mKCC2 transport activity. Unprocessed original blot images for the protein expression are provided in Supplementary Fig. 19.

**$^{86}$Rb$^+$ and Cl$^-$ influx studies.** In the experiments of Fig. 4, we used the $^{86}$Rb$^+$ influx functional assay described previously to examine TM6 residues[23]. $^{86}$Rb$^+$ is no longer commercially available in the US and in the experiments of Fig. 3 we measured maximal hNKCC1 activity as net Cl$^-$-influx, using a Cl$^-$-sensing-YFP in Cl$^-$-depleted cells[78]. This series of experiments was conducted in a HEK cell CRISPR-Cas9 NKCC1-knockout line. In the flux experiments, confluent monolayers of cells in a 96-well plate were pre-incubated for 90 min in a medium containing 140 mM N-methyl glucamine gluconate without Na$^+$, K$^+$, or Cl$^-$—this procedure fully depletes intracellular chloride, which both activates NKCC1 and provides a maximum-YFP-fluorescence reference point. Fluorescence was monitored in each well for 1 min, during which the solution was rapidly changed to the 135 mM Na$^+$, 5 mM K$^+$, 140 mM Cl$^-$ flux medium with or without 100 μM bumetanide—the initial rate of chloride influx was calculated from the initial rate of fluorescence quenching[78]. This method has a much lower background flux rate compared to $^{86}$Rb$^+$ influx measurements, allowing detection of bumetanide-sensitive transport rates that are <1% of the rate in wild-type NKCC1-transfected cells.

**Crosslinking with Cu [1,10-phenanthroline].** Cupric phenanthroline crosslinking assay was carried out as previously described[23]. Briefly, cells were washed with regular medium (135 mM Na$^+$, 5 mM K$^+$,140 mM Cl$^-$, 1 mM Ca$^{2+}$, 1 mM Mg$^{2+}$, 1 mM SO$_4^{2-}$, 1 mM PO$_4^{2-}$, and 5 mM N-methyl glucamine HEPES, pH 7.4.) for 30 min and then exposed to regular medium + 1.5 mM cupric phenanthroline for 10–40 min. 1.5 mM Cupric phenanthroline was made immediately prior to addition from freshly prepared stock solutions of 0.15 M CuSO$_4$ in water and 1.13 M

1,10-phenanthroline in DMSO. In other experiments, cells were pre-incubated in either regular medium or hypotonic low chloride medium (3 mM Cl$^-$ medium diluted 2-fold with water) for 30 min followed by crosslinking with 1.5 mM cupric phenanthroline in either regular, hypotonic low chloride, 0 mM K$^+$ or hypotonic low chloride medium with 250 μM bumetanide. The crosslinking reaction was stopped by the addition of 20 mM EDTA + 20 mM N-ethyl maleimide in a regular medium. Cells were lysed with 100 μl lysis buffer (regular medium containing 20 mM N-ethyl maleimide + 1% Triton-X-100 with protease inhibitor (Complete; Roche Applied Science)) for 15 min, and centrifuged at 14,000 rpm for 15 min, and 10 μl of the supernatant was loaded onto 7.5% Tris-glycine gels. After gel electrophoresis and transfer, membranes were probed with the T4 primary antibody (Developmental Studies Hybridoma Bank) and an IRDye 800 CW goat anti-mouse IgG (LI-COR, Lincoln, NE), and images were acquired using the LI-COR Odyssey imager.

**Crosslinking with Iodine and MTS-3-MTS.** Cells were pre-incubated for 30 min in a regular medium and then treated with 1 mM I$_2$ (freshly diluted from a fresh 500 mM solution in ethanol) in a regular medium for 2 min. For MTS-3-MTS crosslinking, cells were treated with 200 μM MTS-3-MTS (1,3-propanediyl bis-methanethiosulfonate) or 3 mM MTSET (control) (TRC, Ontario CA) in a regular medium for 10 min. The crosslinking reaction was quenched by adding 20 mM EDTA + 20 mM N-ethyl maleimide as above. Unprocessed original blot images of crosslinking studies are provided in Supplementary Figs. 20, 21 and Supplementary Data 3.

**Molecular dynamics simulations and ion-binding site identification for hNKCC1 and mKCC2.** To gain further mechanistic insight into the ion translocation process for hNKCC1 and mKCC2, we employed all-atom molecular simulations (MDs) combining with a well-tempered metadynamics method to explore the possible ion translocation pathway and potential ion-binding sites on their transmembrane (TM) domains. Well-tempered metadynamics[91,92] is an efficient enhanced sampling technique that accumulates the history-dependent Gaussian potential on the collective variable of interest, allowing the simulated biological systems to visit a broader conformational space. In addition to well-tempered metadynamics, the multiple walkers method[93], was also used to enable more efficient exploration of the ion translocation process taking advantage of cluster computing resources. All-atom molecular dynamics simulations were carried out using the single TM domain of hNKCC1 and mKCC2 based on our cryo-EM models. The missing residues and side chains were complemented using the MODELLER package. The CHARMM[94]-GUI web server was used to embed the TM domains of hNKCC1 and mKCC2 into lipid bilayers formed with 1:1 POPC: POPE and 20% cholesterol, consisting of 40 POPE, 40 POPC lipids, and 20 cholesterol molecules in each leaflet, to mimic the membrane environments of hNKCC1 and mKCC2. Histidine residues were modeled in the delta nitrogen tautomeric state and Arg/Lys residues were held positive while the Glu/Asp residues were held negative. 100 mM K$^+$, 100 mM Na$^+$, and 200 mM Cl$^-$ were used to neutralize the MD simulation system of hNKCC1 while 100 mM K$^+$ and 100 mM Cl$^-$ were used to maintain electrical neutrality in the mKCC2 simulation system. The CHARMM36 force fields for proteins, lipids, and ions were implemented and the TIP3P water model was used to dissolve the systems, yielding final systems containing a total of 97,314 atoms for hNKCC1 and 98,172 atoms for mKCC2.

The simulation procedures for hNKCC1 and mKCC2 were similar. After initial energy minimization, each system was heated to 320 K from 0 K over 1 ns in the NVT ensemble using a Berendsen thermostat with 1 fs time step. Harmonic restraints of 5 kcal/mol/Å$^2$ were exerted on the no-hydrogen atoms of proteins and lipids during the minimization and heating stages. After that, semi-isotropic pressure coupling with a Monte Carlo barostat was employed to equilibrate the density of each system in NPT ensemble for 2 ns under 1 bar at 320 K, with harmonic restraints of 5 kcal/mol/Å$^2$ on the backbone atoms of protein and phosphorus atoms of lipid head groups. Harmonic restraints were then gradually reduced to 0.1 kcal/mol/Å$^2$ with intermediate restraints of 2, 1, 0.5, 0.2 kcal/mol/Å$^2$, respectively. Each restraint equilibration was conducted for 2 ns at 320 K and 1 bar using a Langevin thermostat and Monte Carlo barostat in NPT ensemble. Hydrogen mass repartitioning was used to enable a 4-fs time step and all bond lengths involving hydrogen atoms were constrained using the SHAKE algorithm. Periodic boundary conditions were employed and the short-range interactions were cutoff at 1.2 nm while the long-range electrostatic interactions were calculated using the particle-mesh–Ewald method. After the equilibration stage, a further 200 ns relaxation simulation was performed using PMEMD module of Amber18 package with GPU acceleration, prior to performing multiple walkers well-tempered Metadynamics.

The multiple walkers well-tempered Metadynamics simulations (MWs-WTMetaD) were conducted using a GPU-accelerated OpenMM molecular simulation engine combining with PLUMED free energy calculation library. The same general simulation parameters were used for the unbiased MD simulations. After 200 ns unbiased MD simulations, 20 walkers well-tempered Metadynamics simulations were initiated, simultaneously. In addition, 1 μs well-tempered metadynamics simulations of each walker was performed, yielding a cumulative biased simulation time of 20 μs for each system. For the hNKCC1 simulation system, the dynamics of 1 Na$^+$, 1 K$^+$, and 2 Cl$^-$ ions were biased along the membrane normal axis in well-tempered metadynamics

simulations and 20 walkers were initiated simultaneously. The membrane center is defined as the geometry center of all phosphorus atoms of lipids and the $z$ component distance differences between the selected ions and membrane center were biased. An upper boundary restraint was also used to confine the biased ion within a distance of 10 Å around the protein center at $XY$ plane. The gaussian height and width in well-tempered metadynamics simulations were set to 2.5 kJ/mol and 0.5 Å, respectively. The bias factor was set to 10 and the Gaussian bias was saved every 20,000 steps. A similar simulation protocol was used for the mKCC2 simulation system, except that the dynamics of one pair of $K^+ + Cl^-$ ions were biased along the membrane normal axis. All well-tempered metadynamics simulations were initially conducted with ions locating at the bulk solvent. Under periodic boundary conditions of our simulations, with biasing forces the ions could access either extracellular side or intracellular site of the TM domain more efficiently along the membrane normal direction. After 1 μs with 20 walkers in well-tempered metadynamics simulations, several potential ion-binding cavities for hNKCC1 and mKCC2 were seen by clustering the spatial density peaks of biased ions, while the corresponding binding cavities were identified by finding the surrounding residues within a distance cutoff of 4 Å nearby the biased ions.

To further investigate the co-occupancy of bound ions in hNKCC1 and mKCC2, regular MD simulations containing all four ions in the TM domain of hNKCC1 (9 × 1000 ns) and all three ions in the TM mKCC2 (9 × 1000 ns) were performed using the GPU version of PMEMD module in the Amber18 package. The proposed primary ion-binding sites in the TM domain of hNKCC1 and mKCC2 were obtained according to the ion-binding sites observed from the cryo-EM structures of DrNKCC and hKCC1, which are also consistent with our well-tempered Metadynamics simulations. The ion mass densities were determined using volmap plugin in VMD using all regular MD simulation trajectories with a resolution of 1 Å.

**Water permeation in hNKCC1**. The one-dimensional water distribution profile projected onto the Z-axis was obtained using the DENSITY command in CPPTRAJ, a post-analysis module in amber tool package. A 2000 nanosecond regular MD simulation trajectory (20,000 snapshots) was used to perform the analysis with a resolution of 0.25 Å/slice. The one-dimensional water count along Z-axis (from −14 Å to 12 Å) was based on the integration using the following equation in which $dz = 0.25$.

$$\int_{-14}^{12} C_w(z_i) dz$$

The $C_w(z_i)$ is the water distribution histogram profile along $Z$ axis, which could be directly obtained using the DENSITY command in CPPTRAJ.

**Statistics and reproducibility**. The sample sizes and statistical analyses used in Tl$^+$ influx assay and $^{86}Rb^+$ and $Cl^-$ influx studies are presented in the legend of each figure and in Supplementary Data 1.

**Reporting summary**. Further information on research design is available in the Nature Research Reporting Summary linked to this article.

## Data availability

All relevant data are available from the corresponding author upon reasonable request. The 3D cryo-electron microscopy density maps and the coordinates of atomic models have been deposited in the Electron Microscopy Data Bank (EMDB) and in the Protein Data Bank (PDB) with accession code EMD-30542 and PDB 7D10 for human NKCC1, and EMD-30543 and PDB 7D14 for mouse KCC2.

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

## Acknowledgements

We thank the Tsinghua University Branch of China National Center for Protein Sciences (Beijing) for providing the cryo-EM facility support. The computation was completed on the Yanglab GPU workstation. We also thank Esther Bashi from Yale University School of Medicine for technical assistance. This work was supported by funds from the National Key R&D Program of China (2017YFA0504600 and 2016YFA0501100), The National Science Fund for Distinguished Young Scholars (31625008), The National Natural Science Foundation of China (21532004 and 31570733, 31700647, and 21933010), and the National Institute of Health (USA, DK017433 and HL128973).

## Author contributions

M.Y. and B.F. directed the study. S.Z., J.Z., T.L., and Y.L. did the protein purification and detergent screening. S.Z., T.L., W.Z., and L.Z. performed EM sample preparation, data collection, and structural determination with help of M.Y.; G.L. designed molecular dynamics simulation strategy. G.L. and Y.Z. performed simulations and analysis; B.F. and K.R. performed influx assays for gating residues. B.F., P.F., and S.S. prepared the Crispr NKCC1-knockout HEK cells, and carried out the TM6 scanning mutation and dimer crosslinking assays; S.Z. performed the $Tl^+$ influx assay for mKCC2. X.M. and H.D. carried out mass spectrometry analysis. M.Y. and B.F. built the model, drew the figures, and wrote the manuscript with the help of S.Z., J.Z., W.Z., and Y.Z. All authors contributed to the discussion of the data and editing the manuscript.

## Competing interests

The authors declare no competing interests.
