## [Peer Review File · Communications Biology]

Reviewers' comments:

Reviewer #1 (Remarks to the Author):

NKCC and KCC transporters play critical roles in many fundamental physiological processes, such as neuronal excitability, ion transport and cell volume regulation. NKCC1 and KCC2 together regulate the intracellular Cl⁻ concentration in excitatory cells, which controls the response of GABA signaling. Therefore, both proteins are very promising targets to treat seizures.

The authors present the cryo-EM structures on human NKCC1 and mouse KCC2, providing the first structural insights into the important KCC2. Interesting, in KCC2, an extra peptide density was found to be in place to block the pore and the CTD adopts a different relative position to the TM domain. Furthermore, MD simulations were carried out to provide insights into the ion binding. Comprehensive functional studies provided validation of the NKCC1 structure and important insights into its function. In general, these findings are quite interesting and timely, potentially providing important insights into how the transport activity is regulated. There are several issues that need to be addressed.

Main comments:

1) Can the N-terminal peptide of KCC2 be unambiguously assigned? Are there any other parts of the protein that can account for the density? It is important to list the evidence to support the assignment.

2) The functional characterizations on NKCC1 are comprehensive. However, no functional studies were presented on KCC2. Given KCC2 structure is the most exciting new finding in this study and the proposed inhibitory peptide is potentially highly significant for understanding the transport mechanism, functional characterizations should be performed to validate the proposed inhibitory peptide and to probe transport mechanism.

3) It is unclear why the published hKCC1 structure was proposed to be in an active state. Is it possible this is only due to the difference between KCC1 and KCC2? This is important for the proposed activation mechanism. If possible, experimental evidence to support the proposed mechanism will help to increase the impact of the work.

Minor comments:

There are several places "data not shown". It is better to show the data.

Reviewer #2 (Remarks to the Author):

This is an excellent study on the structures of two cation-chloride cotransporters, coming from leading laboratories with a very wide, complementary expertise ranging from cryo-EM to physiological analysis of CCC functions. The authors report the structures of human NKCC1 (hNKCC1) and mouse KCC2 (mKCC2) at 3.5 and 3.8 Å resolution, respectively. The study includes a large number of experimental approaches, including probing by mutagenesis critical amino acids related to ion selectivity and gating.

In their discussions, the authors address a large number of topics which are of high importance for researchers in molecular and cellular physiology, and several points are made with regard to the relevance of the present kind of research and data for the neuroscience community. In particular, the authors propose a model on how phosphorylation regulates ion-transport activity. They discuss fundamental and highly interesting aspects of the obtained structure such as dimerization, ion binding and translocation, the position of known human mutations, and how these mutations may interfere with the functions of the cotransporters.

I have a number of comments listed below. However, first I'd like to state that I am not a structural biologist, but a neurobiologist with a lengthy track record in ion regulation in excitable cells, including cardiac and neuroendocrine cells and, in particular, neurons. While I have some

knowledge at the molecular-biophysical level on ion binding and selectivity of transmembrane proteins (channels and carriers), most of my comments will obviously focus my background. Specialist comments on the details of the structural-biological methods and conclusions should therefore come from other referees of this paper. However, I feel that my comments will help the authors to improve several sections of this excellent paper.

A. General points for consideration

I would urge the authors to at least briefly discuss, or comment on, the following topics when revising this ms:

a) Neurobiologists typically use the term "activation" in CCC functions in an ambiguous manner. It refers to enhanced transport capacity, but usually no distinction is made on whether the "activation" is achieved by an increase in the intrinsic turnover rate of ions at the molecular level; or as an increase in membrane expression of a given type of CCCs. For instance, in their pivotal study (doi: 10.1073/pnas.1415126112), Moss and coworkers state that "the activity of mKCC2 is potentiated via phosphorylation of serine 940 (S940)". However, the "activation" in this context is clearly the result of trafficking, not intrinsic activity. This is also consistent with the fact that in the present structural work, this residue is not dealt with at all. Thus, the word "activation" of CCCs in the present experiments is much more precisely defined than in numerous papers by neurobiologists/electrophysiologists.

Here, I also wonder whether there IS any quantitative information on the effects of (de)phosphorylation on the intrinsic turnover rates of KCC2 and NKCC1. This might be worth a note in the Discussion.

b) A number of CCCs have been claimed to transport water in addition to ions. The authors might perhaps consider whether their work would suggest that water transport as reported by Thomas Zeuthen and his colleagues is consistent with active H₂O transport by (any) CCCs. I am convinced that Dr. Forbush is familiar with this work, so that no references are needed here.

c) NEM is very often used to activate KCC2. It would be interesting to learn whether this effect has any relation to the present findings.

d) Finally, the authors might briefly consider the fact that the C-terminus of KCC2 has been implicated as an important factor in neuronal morphogenesis, especially in the development of dendritic spines. This point is relevant when discussing the modes of action of disease mutations of KCC2 (Ref 65 in the ms).

B. Comments in a page-by-page manner

- Lines 54-57: The (frequently made) statement that "the balance between NKCC and KCC activity" is true only during electrical quiescence in vitro. In the living brain, neurons have subcellular domains with heterogenous functional expression levels of NKCC1 and KCC2, and the major ionic loads arise from activity-dependent channel-mediated fluxes of Cl⁻. I think it would be best to go to the very basics of ion-transporter functions and simply state something like "these transporters generate electrochemical gradients across the neuronal plasma membrane while channels dissipate them, and these interactions set the polarity and driving force of GABA_A receptor-mediated currents". (The expression on line 56, "GABA-stimulated polarization", is not accurate).

Please note in the above context that "NKCC and KCC *activity*" as used in refs 4, 5 and 9 is not identical to what "activity" of CCCs means in the present manuscript; see point a) above.

- Lines 66-68. The description of the "developmental switch" is not accurate. (i) There is no solid evidence that neuronal down-regulation of NKCC1 would be involved. (ii) The reference (19) on KCC2 upregulation is not cited by experts in this area, because in that work, KCC2 primers and KCC2 probe for the Southern blot are not described at all. The lack of the specificity of these unknown probes is evident based on the erroneous observation that KCC2 is expressed in DRG neurons, which is definitely devoid of KCC2. Please note that KCC2 is expressed in central but not peripheral neurons. This is shown in ref 20, which also demonstrates causality (and not mere correlation) between KCC2 upregulation and the depolarizing to hyperpolarizing GABA shift.

Also, "birth" is not a unique reference point for brain development in mammals. In mice and rats, the hyperpolarizing GABA shift takes place after birth, in e.g. guinea pigs and in humans already during prenatal development.

- Line 77: please delete "prevent Cl⁻ loss" which is a clear error.

- Lines 88-89: Increase Cl_i has been demonstrated in several diseases, but whether this is a disease-promoting mechanism itself is currently a hot topic of debate. For instance in epilepsy, the

loss of KCC2 and the consequent increase in Cl⁻ in some neurons may well reflect and adaptive mechanism at the cellular level. So far, there is no evidence that enhancing KCC2 "activity" (whatever the definition here is) would be advantageous in any neurological disease. In some studies, KCC2 blockers have been shown to suppress seizures. I would suggest deleting the very specific arguments and simply state that "changes in Cl⁻ regulation are likely to play a significant role in the molecular etiology of a number of psychiatric and neurological diseases".

The situation with KCC2 mutants in epilepsy (discussed on Lines 424-) is not strictly comparable with down-regulation of KCC2 that takes place after epileptogenesis (cf. section above). However, related to L424- , epilepsy and seizures are currently considered to reflect abnormal neuronal network synchronization rather a simple "hyperexcitability".

- Lines 105-107: "...we provide extensive functional evidence to support hypotheses about ion binding and gating residues and to suggest new hypotheses." This sounds to me a bit too vague as a summary to do justice to the key findings of the present work. Some of the main findings have been summarized in the introductory section of my present statement.

- Lines 157-: KCC2 is the only member of KCC cotransporters that possesses an isotonic transport activity. Other KCCs usually have a very low transport activity at the isotonic conditions, but strongly activated by hypo-osmotic swelling. Would it be possible to speculate more about the structural differences between KCC1 and KCC2 cotransporters due to the presence of the "Iso domain" in the C-terminal part of the KCC2? Could the N terminal inhibitory peptide be somehow involved?

- Lines 179- on the inhibitory N-terminal peptide: Casula et al 2001 (PMID: 11551954) studied the effects of different KCC1 N-terminal truncations on transport activity, none of which showed an increase in the cotransporter activity. On the contrary, a significant decrease was found. This point might deserve a comment in the ms.

According to Markkanen et al 2016 (PMID: 28888841), a SPAK binding site (RFXV/I) is present in the N-termini of all CCC members (at least in one of the N-terminal splice isoforms of each CCC). Is it possible that a direct interaction between SPAK and CCCs could affect the transport activity of the cotransporters by hindering the N-terminal inhibitory peptide from its interaction with the inward-open pore?

- Lines 187-188: The sentence should read "The N-terminus is conserved..., suggesting its..."

- Lines 187- : "N termini are conserved among the KCC family...". This sounds surprising, given the extensive N-terminal heterogeneity of KCC1-4 cotransporters due to the first exons and their alternative splicing. There are at least 4 N-terminal variants for KCC1, 2 variants for KCC2, 7 for KCC3, and 2 for KCC4. Obviously, there is a quite well-conserved part of the N-terminus preceding the TM1, but in general the conservation of the N termini among KCCs is not very high. Also, given the presence of the N-terminal inhibitory peptide, how might this N-terminal heterogeneity affect the regulation of the KCC2 activity in the different splice variants?

- Line 190: "the recent report 57" has not been published yet, it's in available from a pre-submission server.

- Line 213 and elsewhere: Is there any possibility that the mutations studied would have an effect on the inhibitory efficacy of bumetanide?

- Line 439: Again, the "increase in KCC2 activity" most likely reflects changes in protein turnover and trafficking, rather than the intrinsic activation studied in the present work. From a pharmacological point of view, this is obviously a very important distinction.

C. Comments on the amino-acid sequences/numbering

- Unfortunately, there seems to be some confusion with numbering of amino acid residues. In the present work, the cryo-EM structure is solved for the KCC2a isoform, which is 1138 aa long and was published in 2007 (Uvarov et al 2007; PMID: 17715129). Yet, another KCC2 isoform – KCC2b – was cloned much earlier (Payne et al 1996; PMID: 8663311). Please note that KCC2b is by far the more important splice variant in neurons (see Kaila et al., *NatRevNeurosci* 2014; doi: 10.1038/nrn3819), and the KCC2 phosphorylation sites and human mutations have originally been described using the KCC2b-based numbering. Mouse KCC2a and KCC2b isoforms differ by their first exons only (40 unique amino acids in KCC2a versus 17 unique amino acids in KCC2b), thus the same phosphorylation site in the KCC2b isoform (e.g. T906) has a different number (T929) in the KCC2a isoform (23 amino acids shift in the numbering). Moreover, in the Extended Data Fig.17 and elsewhere the positions of human mutations are given using KCC2b aa numbering, but the corresponding positions in the cryo-EM structure are given with reference to KCC2a.

- It is also important to keep in mind that the mouse KCC2a isoform is 1138 aa, while KCC2a

isoforms from the rat, human, and most of the other species are 1139 aa (mouse KCC2 lacks 1 amino acid in the position 1003). Thus, the human disease mutation R952H (original KCC2b notation) corresponds to R975 in the KCC2a notation (23 aa shift between mouse KCC2a and human KCC2b), while another important human mutation R1048W (original KCC2b notation) corresponds to R1070 in mouse KCC2a (a 22 aa shift with regard to the KCC2b isoform).

- There are two splice isoforms also for NKCC1 cotransporter (NKCC1a and NKCC1b) that differ by 16 amino acids in the C-terminal cytoplasmic domain (due to the alternative splicing of the exon-21 in the Slc12a2 gene encoding NKCC1). Using the data on NKCC1 structure (NKCC1a), would it be possible to make any predictions regarding possible differences in the functional properties of the two NKCC1 splice variants?

D. Figure legends

Given the high information content of the figures, I would suggest that the authors include more details in the legends.

What is the alpha-0 designation for the scissor helix in Ext data Fig.15e and f? Alpha-0 is not mentioned in the text and in the Ext data Fig5 alignment.

Reviewer #3 (Remarks to the Author):

This paper describes the cryo-EM structures of human NKCC1 and mouse KCC2 solubilized in detergent. The structural work is well done. A large set of mutagenesis studies have been performed to probe intracellular and extracellular gates and ion binding sites. Molecular dynamics were used to further investigate ion binding. The story is written well and careful attention has been paid to place the work in the context of a large body of literature; it probably is the best among a recent set of CCC structures in this regard.

My major concern is that this is the sixth structural report of CCC family transporters in the past year. In addition to the previous structures of NKCC1 from fish and human, human KCC1, and mouse KCC4, a preprint describing the structures of human KCC2 and KCC3 has been available for several months. The structures presented here are nearly identical to those already reported, although the ion binding sites observed in the other structures are not observed here due to the lower resolution of these data. While the work presented here is well done and is well presented, it is largely confirmatory. The biggest exception is the model presented at the end, which goes further than previous reports to attempt to explain differences in oligomeric assembly and their relation to transporter function. If the authors can further substantiate these models beyond what is presented in the paper it would advance our understanding of CCCs function well beyond what has already been reported. Experiments such as the one described (but not shown) showing an inhibitory effect of W732C mutants that presumably disulfide bridge would support a model for association/dissociation of intracellular domains (and perhaps of dimers into monomers?) in regulating CCC function. Other similar structure-guided tests of the model could also make a compelling case for these hypotheses.

We thank the reviewers for careful and positive critiques of our manuscript -- these suggestions have been very helpful in directing our revision. We have made major changes by a) identifying residues 85-108 of the KCC2 N-terminus as an inhibitory region, b) demonstrating the functional importance of this interaction, c) providing previously “not shown” data of W732C, demonstrating dimer functional interaction, and d) investigating the possibility that NKCC1 could facilitate transmembrane water movement (yes) or is a water-pump (no).

Reviewer #1 (Remarks to the Author):

NKCC and KCC transporters play critical roles in many fundamental physiological processes, such as neuronal excitability, ion transport and cell volume regulation. NKCC1 and KCC2 together regulate the intracellular Cl⁻ concentration in excitatory cells, which controls the response of GABA signaling. Therefore, both proteins are very promising targets to treat seizures.

The authors present the cryo-EM structures on human NKCC1 and mouse KCC2, providing the first structural insights into the important KCC2. Interesting, in KCC2, an extra peptide density was found to be in place to block the pore and the CTD adopts a different relative position to the TM domain. Furthermore, MD simulations were carried out to provide insights into the ion binding.

Comprehensive functional studies provided validation of the NKCC1 structure and important insights into its function. In general, these findings are quite interesting and timely, potentially providing important insights into how the transport activity is regulated. There are several issues that need to be addressed.

Main comments:

1) Can the N-terminal peptide of KCC2 be unambiguously assigned? Are there any other parts of the protein that can account for the density? It is important to list the evidence to support the assignment.

Response: We thank the reviewer for focusing our attention on this issue. With further refinement and application of secondary structure and geometry restraints to examine the cryo-EM density, we have been able to assign the 24 residues in the N-terminus as the interacting peptide. This inhibitory peptide is now shown as Fig. 2a-c. (Page 7, lines 183-186)

2) The functional characterizations on NKCC1 are comprehensive. However, no functional studies were presented on KCC2. Given KCC2 structure is the most exciting new finding in this study and the proposed inhibitory peptide is potentially highly significant for understanding the transport mechanism, functional

characterizations should be performed to validate the proposed inhibitory peptide and to probe transport mechanism.

Response: We have utilized a TI^+ fluorescence influx assay to measure mKCC2 function and examine the role of the interacting peptide region. As now shown in Fig. 2e, f, we find that compared with wild type mKCC2, a delta (85-120) deletion mutation displays increased transporter activity (Fig. 2e, f), strongly supporting an inhibitory role for this region. (Page 8, lines 194-203) [pandemic-related limitations necessitated our use of this third type of flux assay method for these experiments].

3) It is unclear why the published hKCC1 structure was proposed to be in an active state. This is important for the proposed activation mechanism. If possible, experimental evidence to support the proposed mechanism will help to increase the impact of the work.

Response: This question was not directly addressed in the hKCC1 report. The absence of an N-terminal inhibitory peptide in the hKCC1 structure is the strongest argument, in view of the conservation of this clearly inhibitory contact region among family members. We now point this out in proposing that hKCC1 is in an active state.

Is it possible this is only due to the difference between KCC1 and KCC2?

Although possible, this seems highly unlikely. In view of the conservation of the basic structure from bacteria to humans, and the rather “recent” (on a phylogenetic time scale) KCC1-4 splits in vertebrates, a large divergence in a basic structural feature would be unusual. Perhaps more likely is that the hKCC1 structure is an artifact of the hKCC1 construct and preparation, a possibility we point out in the caveats in “Note 1”.

Minor comments:

There are several places “data not shown.” It is better to show the data.

Response: These data are now presented in Fig. 4b and Fig. 6g. Fig. 4b indicates that mutations of S613 and S614 decreased the transport activity. (Page 10, lines 248-251) Fig. 6g indicates the inhibitory effect of W732C mutation, with restoration by DTT. (Page 14, lines 373-375)

Reviewer #2 (Remarks to the Author):

This is an excellent study on the structures of two cation-chloride cotransporters, coming from leading laboratories with a very wide, complementary expertise ranging from cryo-EM to physiological analysis of CCC functions. The authors report the structures of human NKCC1 (hNKCC1) and mouse KCC2 (mKCC2) at 3.5 and 3.8 Å resolution, respectively. The study includes a large number of experimental approaches, including probing by mutagenesis critical amino acids related to ion selectivity and gating.

In their discussions, the authors address a large number of topics which are of high importance for researchers in molecular and cellular physiology, and several points are made with regard to the relevance of the present kind of research and data for the neuroscience community. In particular, the authors propose a model on how phosphorylation regulates ion-transport activity. They discuss fundamental and highly interesting aspects of the obtained structure such as dimerization, ion binding and translocation, the position of known human mutations, and how these mutations may interfere with the functions of the cotransporters.

I have a number of comments listed below. However, first I'd like to state that I am not a structural biologist, but a neurobiologist with a lengthy track record in ion regulation in excitable cells, including cardiac and neuroendocrine cells and, in particular, neurons. While I have some knowledge at the molecular-biophysical level on ion binding and selectivity of transmembrane proteins (channels and carriers), most of my comments will obviously focus my background. Specialist comments on the details of the structural-biological methods and conclusions should therefore come from other referees of this paper. However, I feel that my comments will help the authors to improve several sections of this excellent paper.

A. General points for consideration

I would urge the authors to at least briefly discuss, or comment on, the following topics when revising this ms:

a) Neurobiologists typically use the term “activation” in CCC functions in an ambiguous manner. It refers to enhanced transport capacity, but usually no distinction is made on whether the “activation” is achieved by an increase in the intrinsic turnover rate of ions at the molecular level; or as an increase in membrane expression of a given type of CCCs. For instance, in their pivotal study (doi: 10.1073/pnas.1415126112), Moss and coworkers state that “the activity of mKCC2 is potentiated via phosphorylation of serine 940 (S940)”. However, the “activation” in this context is clearly the result of trafficking, not intrinsic activity. This is also consistent with the fact that in the present structural work, this residue is not dealt with at all. Thus, the word “activation” of CCCs in the present experiments is much more precisely defined than in numerous papers by neurobiologists/electrophysiologists.

Response: We thank reviewer 2 for clarifying comments and very helpful suggestions. This is a very important point and we have tried to be clearer in the revised manuscript. In the revised introduction we point out that regulation by phosphorylation involves both changes in intrinsic transport activity and trafficking and we now mention the activating serine phosphorylation, citing the Moss study. Elsewhere we now use the term “transport activity” to mean “intrinsic transport activity” (as the Moss paper does), and refer to KCC2 trafficking as well when we discuss physiology. [the Moss paper uses “KCC2 activity” to refer to the net result of intrinsic changes and trafficking, but it uses “transport activity” to refer to the intrinsic changes, eg: “Recent evidence indicates that the transport activity and membrane trafficking of KCC2 are modulated by protein kinase C (PKC)-dependent phosphorylation of residue S940 within its cytoplasmic domain”].

[It is our working hypothesis that for KCC2 (and NKCC2) phosphorylation affects both intrinsic activity and trafficking simultaneously, and available evidence generally supports this (this is challenging to study since intrinsic activity is hard to study when the protein has not reached the membrane). It is clear that trafficking is a major component of KCC2 regulation but for other KCCs it is not so clear, and reports of intrinsic activity changes could be contaminated by transporter internalization. However, early human red cell experiments clearly demonstrated changes in intrinsic K-Cl cotransporter (KCC3) activity related to phosphorylation, in a system where membrane trafficking was unlikely. We suspect that the basic mechanisms are conserved among family members and that cellular machinery plays a big role in the specific physiological manifestation.]

Here, I also wonder whether there is any quantitative information on the effects of (de)phosphorylation on the intrinsic turnover rates of KCC2 and NKCC1. This might be worth a note in the Discussion.

Response: For NKCC1, the transporter goes from undetectable flux to about 3500 turnovers/s upon phosphorylation at 37 °C. We are not aware of a published turnover rate for KCCs. A discussion of these points seems suited for a review and beyond the scope of the present manuscript.

b) A number of CCCs have been claimed to transport water in addition to ions. The authors might perhaps consider whether their work would suggest that water transport as reported by Thomas Zeuthen and his colleagues is consistent with active H₂O transport by (any) CCCs. I am convinced that Dr. Forbush is familiar with this work, so that no references are needed here.

Response: We thank the reviewer for suggesting this -- to address this point, we have added Supplementary Fig. 14 and a section in Results. The results demonstrate the possibility of water permeability through NKCC1, but the low number of water molecules rules out the isosmotic water-pump hypothesis. [This hypothesis was problematic from the outset: a sphere of 600 water molecules would be 32 angstroms in diameter.] (Page 13, lines 341-358)

c) NEM is very often used to activate KCC2. It would be interesting to learn whether this effect has any relation to the present findings.

Response: NEM is a membrane-permeant non-specific sulfhydryl-modifying reagent that activates KCC and inactivates NKCC. Although NEM does have direct effects on CCCs, the activating effect on KCC was shown by Michael Jennings to be through inhibition of the inactivating kinase (Jennings and Schulz, J.Gen.Physiol., 1991, PMC2216490). This explanation has been further supported and extended, most recently this year (Zhang et al, PlosOne 2020, PMC7228128). In view of the non-specific and primarily indirect NEM actions, we have not commented.

d) Finally, the authors might briefly consider the fact that the C-terminus of KCC2 has been implicated as an important factor in neuronal morphogenesis, especially in the development of dendritic spines. This point is relevant when discussing the modes of action of disease mutations of KCC2 (Ref 65 in the ms).

Response: We have introduced the importance of KCC2 in neuronal morphogenesis in the discussion of disease mutations and added an additional reference. (Page 17, lines 456-457) We are not able to make specific predictions because the previously identified residue (R952) is in a region not resolved in the structure.

B. Comments in a page-by-page manner

*- Lines 54-57: The (frequently made) statement that “the balance between NKCC and KCC activity” is true only during electrical quiescence in vitro. In the living brain, neurons have subcellular domains with heterogenous functional expression levels of NKCC1 and KCC2, and the major ionic loads arise from activity-dependent channel-mediated fluxes of Cl⁻. I think it would be best to go to the very basics of ion-transporter functions and simply state something like “these transporters generate electrochemical gradients across the neuronal plasma membrane while channels dissipate them, and these interactions set the polarity and driving force of GABAA receptor-mediated currents”. (The expression on line 56, “GABA-stimulated polarization”, is not accurate). Please note in the above context that “NKCC and KCC *activity*” as used in refs 4, 5 and 9 is not identical to what “activity” of CCCs means in the present manuscript; see point a) above.*

Response: We have corrected the statement in the revised manuscript as suggested. (Page 3, lines 53-55)

- Lines 66-68. The description of the “developmental switch” is not accurate. (i) There is no solid evidence that neuronal down-regulation of NKCC1 would be involved. (ii) The reference (19) on KCC2 upregulation is not cited by experts in this area, because in that work, KCC2 primers and KCC2 probe for the Southern blot are not described at all. The lack of the specificity of these unknown probes is evident based on the erroneous observation that KCC2 is expressed in DRG neurons, which is definitely devoid of KCC2. Please note that KCC2 is expressed in central but not peripheral neurons. This is shown in ref 20, which also demonstrates causality (and not mere correlation) between KCC2 upregulation and the depolarizing to hyperpolarizing GABA shift. Also, “birth” is not a unique reference point for brain development in mammals. In mice and rats, the hyperpolarizing GABA shift takes place after birth, in e.g. guinea pigs and in humans already during prenatal development.

Response: We improved this presentation by removing NKCC1 from the switch, removing ref. 19, and removing the fixed reference of “birth”. (Page 3, lines 65-66)

- Line 77: please delete “prevent Cl⁻ loss” which is a clear error.

Response: Deleted.

- Lines 88-89: Increase Cl_i has been demonstrated in several diseases, but whether this is a disease-promoting mechanism itself is currently a hot topic of debate. For instance, in epilepsy, the loss of KCC2 and the consequent increase in Cl_i in some neurons may well reflect and adaptive mechanism at the cellular level. So far, there is no evidence that enhancing KCC2 “activity” (whatever the definition here is) would be advantageous in any neurological disease. In some studies, KCC2 blockers have been shown to suppress seizures. I would suggest deleting the very specific arguments and simply state that “changes in Cl⁻ regulation are likely to play a significant role in the molecular etiology of a number of psychiatric and neurological diseases”.

Response: In revision we have used this more general statement as suggested, although we retain some of the discussion of the Cl_i hypothesis.

The situation with KCC2 mutants in epilepsy (discussed on Lines 424-) is not strictly comparable with down-regulation of KCC2 that takes place after epileptogenesis (cf. section above). However, related to L424- , epilepsy and seizures are currently considered to reflect abnormal neuronal network synchronization rather a simple “hyperexcitability”.

Response: We have replaced “hyperexcitability” with “abnormal neuronal network synchronization”. (Page 17, lines 454-455)

- Lines 105-107: “...we provide extensive functional evidence to support hypotheses about ion binding and gating residues and to suggest new hypotheses.” This sounds to me a bit too vague as a summary to do justice to the key findings of the present work. Some of the main findings have been summarized in the introductory section of my present statement.

Response: We have revised this to be more specific. (Page 5, lines 106-109)

- Lines 157-: KCC2 is the only member of KCC cotransporters that possesses an isotonic transport activity. Other KCCs usually have a very low transport activity at the isotonic conditions, but strongly activated by hypo-osmotic swelling. Would it be possible to speculate more about the structural differences between KCC1 and KCC2 cotransporters due to the presence of the “Iso domain” in the C-terminal part of the KCC2? Could the N terminal inhibitory peptide be somehow involved?

Response: The “Iso domain” is not resolved in our structure, but even if it were we think it would be difficult to speculate on mechanism, as interaction with (or identity of) the cellular volume-regulatory machinery is still a complete mystery.

- Lines 179- on the inhibitory N-terminal peptide: Casula et al 2001 (PMID: 11551954) studied the effects of different KCC1 N-terminal truncations on transport activity, none of which showed an increase in the cotransporter activity. On the contrary, a significant decrease was found. This point might deserve a comment in the ms.

Response: In revision we briefly discuss this result, which we presume is due to necessity for something further upstream in the N-terminus for correct folding or function.

According to Markkanen et al 2017 (PMID: 28888841), a SPAK binding site (RFXV/I) is present in the N-termini of all CCC members (at least in one of the N-terminal splice isoforms of each CCC). Is it possible that a direct interaction between SPAK and CCCs could affect the transport activity of the cotransporters by hindering the N-terminal inhibitory peptide from its interaction with the inward-open pore?

Response: This is a fascinating question and entirely possible, but way beyond the scope of this investigation. A similar situation occurs in NKCC with SPAK and PP1

binding sites as well as phosphoacceptor sites all in the N-terminus, but even with very many papers on the individual sites, interactions have been suspected but never really addressed.

- Lines 187-188: *The sentence should read “The N-terminus is conserved..., suggesting its...” - Lines 187- : “N termini are conserved among the KCC family...”. This sounds surprising, given the extensive N-terminal heterogeneity of KCC1-4 cotransporters due to the first exons and their alternative splicing. There are at least 4 N-terminal variants for KCC1, 2 variants for KCC2, 7 for KCC3, and 2 for KCC4. Obviously, there is a quite well-conserved part of the N-terminus preceding the TM1, but in general the conservation of the N termini among KCCs is not very high. Also, given the presence of the N-terminal inhibitory peptide, how might this N-terminal heterogeneity affect the regulation of the KCC2 activity in the different splice variants?*

Response: This section has been revised to describe conservation of the specific region of the N-terminus and the interacting residues. (Page 8, lines 191-193)

- Line 190: *“the recent report 57” has not been published yet, it’s in available from a pre-submission server.*

Response: We have deleted this reference.

- Line 213 and elsewhere: *Is there any possibility that the mutations studied would have an effect on the inhibitory efficacy of bumetanide?*

Response: Yes, we have pursued some of the elevated “+bumetanide” values shown in the figures. The result, a finding of markedly lower bumetanide affinity in two mutants, is now shown in Supplementary Fig. 10c and discussed briefly in the results. (Page 9, lines 226-229)

- Line 439: Again, the “increase in KCC2 activity” most likely reflects changes in protein turnover and trafficking, rather than the intrinsic activation studied in the present work. From a pharmacological point of view, this is obviously a very important distinction.

Response: Revised: “to increasing intrinsic KCC2 activity and membrane trafficking” (Page 18, lines 470)

C. Comments on the amino-acid sequences/numbering

- Unfortunately, there seems to be some confusion with numbering of amino acid residues. In the present work, the cryo-EM structure is solved for the KCC2a isoform, which is 1138 aa long and was published in 2007 (Uvarov et al 2007; PMID: 17715129). Yet, another KCC2 isoform – KCC2b – was cloned much earlier (Payne et al 1996; PMID: 8663311). Please note that KCC2b is by far the more important splice variant in neurons (see Kaila et al., NatRevNeurosci 2014; doi: 10.1038/nrn3819), and the KCC2 phosphorylation sites and human mutations have originally been described using the KCC2b-based numbering. Mouse KCC2a and KCC2b isoforms differ by their first exons only (40 unique amino acids in KCC2a versus 17 unique amino acids in KCC2b), thus the same phosphorylation site in the KCC2b isoform (e.g. T906) has a different number (T929) in the KCC2a isoform (23 amino acids shift in the numbering). Moreover, in the Extended Data Fig.17 and elsewhere the positions of human mutations are given using KCC2b aa numbering, but the corresponding positions in the cryo-EM structure are given with reference to KCC2a.

- It is also important to keep in mind that the mouse KCC2a isoform is 1138 aa, while KCC2a isoforms from the rat, human, and most of the other species are 1139 aa (mouse KCC2 lacks 1 amino acid in the position 1003). Thus, the human disease mutation R952H (original KCC2b notation) corresponds to R975 in the KCC2a notation (23 aa shift between mouse KCC2a and human KCC2b), while another important human mutation R1048W (original KCC2b notation) corresponds to R1070 in mouse KCC2a (a 22 aa shift with regard to the KCC2b isoform).

Response: We thank the reviewer for pointing out the tangle of numbers. In revision we note at the first occurrence on line 119 that we are studying hNKCC1a and mKCC2a, and we think it implicit that we use those throughout the manuscript, with the exception of discussion of human disease where we now state that the numbering is according to hNKCC2b. Supplementary Fig. 18 now lists both numberings for the human disease mutations.

- There are two splice isoforms also for NKCC1 cotransporter (NKCC1a and NKCC1b) that differ by 16 amino acids in the C-terminal cytoplasmic domain (due to the alternative splicing of the exon-21 in the Slc12a2 gene encoding NKCC1). Using the data on NKCC1 structure (NKCC1a), would it be possible to make any predictions regarding possible differences in the functional properties of the two NKCC1 splice variants?

Response: This splice region has been described by one of us (Carmosino...Forbush, 2008) to be critical in the membrane sorting of NKCC1 vs NKCC2; but there were no obvious functional differences attributable to the splice domain in that work or in

early reports from other labs. The region is not well conserved and is not resolved in our structure.

D. Figure legends

Given the high information content of the figures, I would suggest that the authors include more details in the legends.

What is the alpha-0 designation for the scissor helix in Ext data Fig.15e and f? Alpha-0 is not mentioned in the text and in the Ext data Fig5 alignment.

Response: alpha-0 is now explained in the legend, and some legends have been expanded with greater detail.

Reviewer #3 (Remarks to the Author):

This paper describes the cryo-EM structures of human NKCC1 and mouse KCC2 solubilized in detergent. The structural work is well done. A large set of mutagenesis studies have been performed to probe intracellular and extracellular gates and ion binding sites. Molecular dynamics were used to further investigate ion binding. The story is written well and careful attention has been paid to place the work in the context of a large body of literature; it probably is the best among a recent set of CCC structures in this regard.

My major concern is that this is the sixth structural report of CCC family transporters in the past year. In addition to the previous structures of NKCC1 from fish and human, human KCC1, and mouse KCC4, a preprint describing the structures of human KCC2 and KCC3 has been available for several months. The structures presented here are nearly identical to those already reported, although the ion binding sites observed in the other structures are not observed here due to the lower resolution of these data. While the work presented here is well done and is well presented, it is largely confirmatory. The biggest exception is the model presented at the end, which goes further than previous reports to attempt to explain differences in oligomeric assembly and their relation to transporter function. If the authors can further substantiate these models beyond what is presented in the paper it would advance our understanding of CCCs function well beyond what has already been reported.

Experiments such as the one described (but not shown) showing an inhibitory effect of W732C mutants that presumably disulfide bridge would support a model for association/dissociation of intracellular domains (and perhaps of dimers into monomers?) in regulating CCC function. Other similar structure-guided tests of the model could also make a compelling case for these hypotheses.

Response: We thank the reviewer for the positive suggestions. We have added the W732C data as Fig 6g (included above), including the effect of DTT in restoring activity – these results strongly support involvement of dimer interaction in CCC regulation. Supporting another part of the model, the identification of the conserved inhibitory region in the N-terminus of KCC2 and the structure-guided test of its function makes a compelling case for participation of the KCC N-terminus in regulation of that transporter.

REVIEWERS' COMMENTS:

Reviewer #1 (Remarks to the Author):

Overall, the authors have adequately addressed my comments. My only remaining suggestion is to tune down the proposed activation model since the activation states of structures cannot be firmly established. Possibly, it might be better move Fig. 7 into supplementary figures.

Reviewer #2 (Remarks to the Author):

The authors have done an excellent job when revising the paper. The addition of novel data including the absence of active water transport by NKCC1 will clarify some long-debated questions; and the data on bumetanide binding are of immediate value for transport physiologists using e.g. mutated constructs.

I have only minor comments.

1. References #9 (which appeared in 2000) and #18 (year 1997) are outdated, and they have in fact lots of major conclusions that have been refuted. Thus, the reader has to look for more recent reviews (not available in the list of refs) to put the data of the present work into context. The currently most comprehensive and widely cited review on neuronal CCCs is the one by Kaila et al in *NatRevNeurosci* 2014 (doi:10.1038/nrn3819).

2. The sentence on lines 54-56 still needs some minor changes (please add "Cl-" and "chloride" as indicated: "These transporters generate electrochemical Cl- gradients..., ... and driving force for GABAA receptor-mediated chloride currents").

The latter "chloride" addition is important, because in some neurons with a very negative resting potential (e.g., adult neocortical pyramids at rest), most of the GABA_AR-mediated current is carried by HCO₃⁻ (lower permeability but much larger driving force than for Cl⁻).

Here, please also note comment 1. on relevant references.

3. Lines 65-67. KCC2 is expressed in central (CNS) neurons only. Moreover, there is really no "switch" but rather a smooth developmental "shift" which takes place during neuronal maturation (not "in the adult brain"). Therefore, please rephrase sentence to something like:

Line 65: "NKCC1 and KCC2 are expressed in mammalian central neurons, in which a developmental hyperpolarizing shift in the action of GABA takes place during neuronal maturation because of an increase in the functional expression of KCC2 (refs)".

Please note that the idea that the above developmental shift is associated with (or caused by) down-regulation of NKCC1 has been refuted (and notably, recent work shows that by far most of the NKCC1 in brain tissue is expressed in glial cells, making NKCC1/KCC2 ratios meaningless).

4. Lines 67- 68: It seems that because of my oversight in the previous statement, it is stated here that GABA is excitatory in adult sensory neurons. This is not the case, and the well-known primary afferent depolarization is functionally inhibitory. Moreover, sensory neurons are not part of the CNS, and they never express KCC2. Therefore, please rephrase according to the following:

"Thus, by raising [Cl]_i, NKCC1 is responsible for the depolarizing and sometimes excitatory GABA actions in immature central neurons, and also for the GABA-mediated (but functionally inhibitory) depolarization in adult sensory neurons, while the CNS-specific KCC2... "

All the above details have been covered in the review provided above (point 1).

5. Line 402: "Structure of the Regulatory Cytosolic Domain of a Eukaryotic Potassium-Chloride cotransporter", by Zimanyi et al 2020, *Structure*, should be cited (<https://doi.org/10.1016/j.str.2020.06.009>).

6. Lines 415-416. Am I right in that the dimeric "condensed" state with tightly interacting C-terms is rather an inactive KCC2 state? Does it mean that the SDS-resistant dimers are more easily formed from these "swapped" KCC2 subunits, i.e inactive KCC2 cotransporter?

Typo on line 196: "found that *of* deletion of most of..."

Reviewer #3 (Remarks to the Author):

I have no further comments or suggestions. The authors have addressed concerns raised in a satisfactory way.

REVIEWERS' COMMENTS:

We thank the editor and the reviewers for their careful evaluations and constructive comments, which helps to improve the scientific implications of our study. We have now addressed the minor comments by the reviewers.

Reviewer #1:

Remarks to the Author:

Overall, the authors have adequately addressed my comments. My only remaining suggestion is to tune down the proposed activation model since the activation states of structures cannot be firmly established. Possibly, it might be better move Fig. 7 into supplementary figures.

Response: We thank the reviewer for the helpful suggestions and we have revised the manuscript by making changes in the first paragraph of discussion and within the figure legend to tune down the proposed activation model. We kept Fig. 7 in place since we feel that this main point has been tuned down in the discussion section and is appropriately placed with the main figures.

Reviewer #2:

Remarks to the Author:

The authors have done an excellent job when revising the paper. The addition of novel data including the absence of active water transport by NKCC1 will clarify some long-debated questions; and the data on bumetanide binding are of immediate value for transport physiologists using e.g. mutated constructs.

I have only minor comments.

1. References #9 (which appeared in 2000) and #18 (year 1997) are outdated, and they have in fact lots of major conclusions that have been refuted. Thus, the reader has to look for more recent reviews (not available in the list of refs) to put the data of the present work into context. The currently most comprehensive and widely cited review on neuronal CCCs is the one by Kaila et al in NatRevNeurosci 2014 (doi:10.1038/nrn3819).

Response: We would like to thank the reviewer for the ongoing and enthusiastic support on our study. In the revised manuscript, we have deleted the outdated references (previous #9 and #18) and added a comprehensive and widely cited review (Kaila et al in NatRevNeurosci 2014)

2. The sentence on lines 54-56 still needs some minor changes (please add “Cl-“ and “chloride” as indicated: “These transporters generate electrochemical Cl-

gradients..., ... and driving force for GABAA receptor-mediated chloride currents”.

The latter “chloride” addition is important, because in some neurons with a very negative resting potential (e.g., adult neocortical pyramids at rest), most of the GABA_AR-mediated current is carried by HCO₃⁻ (lower permeability but much larger driving force than for Cl⁻).

Here, please also note comment 1. on relevant references.

Response: We have rephrased the sentence as suggested.

3. Lines 65-67. KCC2 is expressed in central (CNS) neurons only. Moreover, there is really no “switch” but rather a smooth developmental “shift” which takes place during neuronal maturation (not “in the adult brain”). Therefore, please rephrase sentence to something like:

Line 65: “NKCC1 and KCC2 are expressed in mammalian central neurons, in which a developmental hyperpolarizing shift in the action of GABA takes place during neuronal maturation because of an increase in the functional expression of KCC2 (refs)”.

Please note that the idea that the above developmental shift is associated with (or caused by) down-regulation of NKCC1 has been refuted (and notably, recent work shows that by far most of the NKCC1 in brain tissue is expressed in glial cells, making NKCC1/KCC2 ratios meaningless).

Response: We have rephrased the description as suggested.

4. Lines 67- 68: It seems that because of my oversight in the previous statement, it is stated here that GABA is excitatory in adult sensory neurons. This is not the case, and the well-known primary afferent depolarization is functionally inhibitory. Moreover, sensory neurons are not part of the CNS, and they never express KCC2. Therefore, please rephrase according to the following:

“Thus, by raising [Cl]_i, NKCC1 is responsible for the depolarizing and sometimes excitatory GABA actions in immature central neurons, and also for the GABA-mediated (but functionally inhibitory) depolarization in adult sensory neurons, while the CNS-specific KCC2... “

All the above details have been covered in the review provided above (point 1).

Response: We have rephrased the sentence as suggested.

5. Line 402: "Structure of the Regulatory Cytosolic Domain of a Eukaryotic Potassium-Chloride cotransporter", by Zimanyi et al 2020, Structure, should be cited (<https://doi.org/10.1016/j.str.2020.06.009>).

Response: We have added the new reference in the revised manuscript.

6. Lines 415-416. Am I right in that the dimeric "condensed" state with tightly interacting C-terms is rather an inactive KCC2 state? Does it mean that the SDS-resistant dimers are more easily formed from these "swapped" KCC2 subunits, i.e. inactive KCC2 cotransporter?

Response: (a): yes, the dimeric "condensed" state with tightly interacting C-terms is proposed to be an inactive KCC2 state, as described in the first paragraph of discussion. (b): SDS-resistant dimers are well outside the scope of this paper; it is clear that for NKCC, SDS-resistance in dimer formation is acquired during sample preparation and there is no way to know if reflects the state before the addition of SDS, so a "yes" answer would be purely speculative.

Typo on line 196: "found that *of* deletion of most of..."

Response: We have deleted the typo as suggested.

Reviewer #3:

Remarks to the Author:

I have no further comments or suggestions. The authors have addressed concerns raised in a satisfactory way.

Response: We thank the reviewer for the effort in evaluating the manuscript and we are grateful for the valuable suggestions during the review process.